# H2A.Z deposition by the SWR complex is stimulated by polyadenine DNA sequences in nucleosomes

Cynthia Converso[1]◉, Leonidas Pierrakeas[1]◉, Lirong Chan[1], Shalvi Chowdhury[2], Emily de Onis[2], Vyacheslav I. Kuznetsov[3,4], John M. Denu[3,4], Ed Luk◉[1,2]*

1 Department of Biochemistry and Cell Biology, Stony Brook University, Stony Brook, New York, United States of America, 2 Renaissance School of Medicine, Stony Brook University, Stony Brook, New York, United States of America, 3 Department of Biomolecular Chemistry, School of Medicine and Public Health, University of Wisconsin, Madison, Wisconsin, United States of America, 4 Wisconsin Institute for Discovery, University of Wisconsin, Madison, Wisconsin, United States of America

◉ These authors contributed equally to this work.
* ed.luk@stonybrook.edu

**Data availability statement:** Sequencing data are available from the NIH Sequence Read Archive (SRA) under accession number PRJNA1060626 and from the Gene Expression

## Abstract

The variant histone H2A.Z is deposited into nucleosomes immediately downstream of promoters, where it plays a critical role in transcription. The site-specific deposition of H2A.Z is catalyzed by the SWR complex, a conserved chromatin remodeler with affinity for promoter-proximal nucleosome-depleted regions (NDRs) and histone acetylation. By comparing the genomic distribution of H2A.Z in wild-type and SWR-deficient cells, we found that SWR is also responsible for depositing H2A.Z at thousands of non-canonical sites not directly linked to NDRs or histone acetylation. To understand the targeting mechanism of H2A.Z, we presented SWR to a library of canonical nucleosomes isolated from yeast and analyzed the preferred substrates. Our results revealed that SWR preferentially deposited H2A.Z into a subset of endogenous H2A.Z sites, which are overrepresented by polyadenine tracts on the top strands of the DNA duplex at the nucleosomal entry-exit sites. Insertion of polyadenine sequences into recombinant nucleosomes near the outgoing H2A-H2B dimer enhanced SWR's affinity for the nucleosomal substrate and increased its H2A.Z insertion activity. These findings suggest that the genome encodes sequence-based information that facilitates remodeler-mediated targeting of H2A.Z.

## Introduction

Nucleosomes are the building blocks of chromosomes in a eukaryotic cell [1]. A typical nucleosome has a protein core made up of eight histones, two copies of histone H2A, H2B, H3 and H4, that is coiled around by 146-basepair (bp) DNA in 1.65 superhelical turns [2]. Individual nucleosomes are connected by linker DNA of variable lengths and are positioned along genes in a non-random, stereotypic pattern that is functionally important for transcription [3]. In yeast, a nucleosome-depleted region (NDR), 80-200 bp in length, is typically associated with a promoter [4]. Flanking the NDR are the +1 and -1 nucleosomes. In yeast, the upstream edge of a +1 nucleosome overlaps the transcription start site (TSS) [5]. Downstream of the +1

Omnibus (GEO) under access number GSE252824. Custom Python scripts are deposited in the Zenodo database under https://doi.org/10.5281/zenodo.14807456.

**Funding:** This work was supported by the National Institutes of Health (NIH) grants R01GM104111, R01GM147795, and R35GM156364 to E.L. and R35 GM149279 to J.M.D. and the NIH Shared Instrumentation Grant (SIG) S10OD024986 to the Genomics Core Facility of Stony Brook University Renaissance School of Medicine. The funders had no role in study design, data collection and analysis, decision to publish, or preparation of the manuscript.

**Competing interests:** I have read the journal's policy, and the authors of this manuscript declare the following competing interests: J.M.D. is a co-founder of Galilei BioSciences and a consultant for Evrys Bio. The organizations had no role in study design, data collection and analysis, decision to publish, or preparation of the manuscript.

nucleosome are the +2, +3 nucleosomes and so on, organized into a closely spaced array [4]. Since the nucleosome structure generally occludes DNA elements, focused assembly of the transcriptional preinitiation complex (PIC) occurs at the NDR [6]. Studies in yeast showed perturbations of the position and occupancy of the native chromatin arrangement led to aberrant transcriptional response and initiation from cryptic promoters within genic regions, negatively impacting the fitness of cells [7–10].

The +1 nucleosome is frequently installed with the histone variant H2A.Z, a prominent landmark associated with active and poised promoters [5,11]. H2A.Z is required for rapid transcriptional response in yeast and is essential for life in metazoans [12–15]. RNA polymerase II (Pol II)-mediated transcription preferentially ejects H2A.Z over H2A, suggesting that an H2A.Z-containing nucleosome at +1 is the chromatin state designated for initiation [16,17]. However, the role of H2A.Z appears to extend beyond transcription initiation. In yeast, optimal phosphorylation of Pol II C-terminal domain (CTD), elongation factor recruitment, Pol II elongation rate, and RNA splicing require H2A.Z [18,19]. In metazoans, H2A.Z nucleosomes at +1 sites regulate the pause release of Pol II [20,21]. These studies suggest that H2A.Z functions at a juncture whereby Pol II transitions from initiation to productive elongation.

The site-specific deposition of H2A.Z is carried out by SWR, a 14-component chromatin remodeling complex [22–24]. The 'SWR' nomenclature distinguishes the complex from its core subunit, Swr1, which is a member of the SWI/SNF-related family ATPases [25]. Unlike other chromatin remodelers in the SWI/SNF family, which slide nucleosomes, SWR catalyzes an ATP-driven remodeling reaction that installs H2A.Z into nucleosomes. At the molecular level, SWR replaces the two H2A-H2B (A-B) dimers within a canonical H2A-containing (AA) nucleosome with free H2A.Z-H2B (Z-B) dimers that are delivered by histone chaperones in a stepwise manner [26,27]. As such, SWR produces a heterotypic H2A/H2A.Z (AZ) nucleosome as an intermediate before forming the homotypic H2A.Z (ZZ) nucleosome as the final product [28]. The targeting of SWR is mediated in part by its Swc2 and Swc3 subunits, which contribute to the sensing and binding of the NDR adjacent to +1 nucleosomes [29,30]. Upon binding to the NDR, SWR can flip the nucleosome to insert Z-B dimers on its opposite faces [28,31]. Histone acetylation also contributes to SWR recruitment and H2A.Z deposition [11,30]. SWR bears a tandem bromodomain in the Bdf1 subunit and a YEATS domain in the Yaf9 subunit that are reader modules for acetylated histone tails [11,32,33]. However, SWR's affinity for the NDR and histone acetylation cannot fully explain the genomic distribution of H2A.Z in vivo, as many H2A.Z-containing nucleosomes are located at NDR-distal sites and are not particularly enriched for histone acetylation [5,16,34]. Therefore, our understanding of the mechanisms underlying H2A.Z targeting is incomplete.

The H2A.Z insertion activity of SWR is influenced by the DNA sequences of the nucleosomal substrates. When presented with nucleosomal substrates assembled with the Widom '601' DNA sequence, which is asymmetric on either side of the dyad axis, SWR inserts H2A.Z onto the opposite faces of the nucleosome at different rates [28]. The determinant of the insertion bias was pared down to a 16-bp region in front of the remodeling ATPase engagement site, where the favored sequence has longer poly(dG:dC) tracts (but equal GC content) [28]. However, the sequence variation between the two halves of the Widom sequence is too limiting to fully understand the potential effects that DNA sequence has on the remodeling activity of SWR.

The influence of DNA sequence on chromatin remodeling activities is not limited to the SWR complex but is also observed for other remodelers, such as RSC and Chd1 [35,36]. In the case of RSC, which widens NDRs at promoters, the sequence-specific effect on remodeling activity is biologically significant [37]. RSC is stimulated by poly(dA:dT) sequences, which

are frequently found at promoter-proximal NDRs, suggesting that the genome is encoded with information to direct remodeling activity in a site-specific manner [36,37]. Whether the genome is encoded with information that can fine-tune the H2A.Z deposition activity of SWR is unknown.

In this study, we determined the genomic distribution of SWR-dependent H2A.Z by comparing chromatin-bound H2A.Z in wild type (WT) and *SWC2*-deficient yeast cells. We found that SWR is responsible for depositing H2A.Z not only at NDR-proximal sites, but also at thousands of NDR-distal sites that are not particularly enriched for histone acetylation. This observation motivated us to test the hypothesis that DNA sequences contribute to SWR's substrate specificity across the yeast genome. We developed an in vitro approach that utilizes mono-nucleosome libraries isolated from yeast cells to interrogate the substrate specificity of SWR. We found that poly(dA:dT) enhances the H2A.Z insertion activity of SWR, partly by increasing its affinity for nucleosomal substrates. This finding provides insight into how SWR targets promoter-proximal nucleosomes and certain NDR-distal sites enriched in poly(dA:dT) sequences.

## Results

### SWR deposits H2A.Z predominantly at +1 nucleosomes but also at non-canonical sites

To evaluate SWR's role in global H2A.Z deposition, we compared chromatin-bound H2A.Z levels in WT cells to those in *swc2Δ* mutants, where SWR is inactive for H2A.Z deposition [38]. Chromatin-bound H2A.Z levels were measured using the in vivo disulfide crosslinking method VivosX [39]. This technique utilized cysteine-modified *HTZ1(T46C)* and *HTA1(N39C)* as the sole sources of H2A.Z and H2A, respectively. Because H2A.Z is the only variant H2A in yeast and no other cysteines are present in core histones, the cysteine probes on H2A.Z and/or H2A, positioned at the interdisk interface of nucleosomes, are close enough for disulfide crosslinking (Fig 1A). Crosslinking was initiated with the cell-permeable, thiol-specific crosslinker 4,4′-dipyridyl disulfide (4-DPS). To facilitate resolution of crosslinking adducts, the *HTZ1(T46C)* gene was fused to a C-terminal 2xFLAG tag, and the *HTA1(N39C)* gene was fused to an N-terminal 2xV5 tag. As such, the crosslinked adducts of H2A-to-H2A.Z and H2A.Z-to-H2A.Z infer AZ and ZZ nucleosomes, respectively, while uncrosslinked H2A.Z infers non-nucleosomal H2A.Z (Fig 1B) [39]. Band intensities were normalized to histone H4, which serves as a loading control (Fig 1B and 1C).

In WT cells, we estimated that 12% of bulk H2A.Z is in the ZZ configuration, 57% in the AZ configuration, and 31% in non-nucleosomal forms (Fig 1B and 1C). In *swc2Δ*, ZZ nucleosomes decreased to 2.5%, AZ nucleosomes decreased to 34%, while non-nucleosomal H2A.Z increased to 47% (Fig 1B and 1C). Similarly, deletion of the *SWR1* gene, which encodes SWR's ATPase motor, resulted in a comparable reduction in ZZ and AZ nucleosomes as observed in *swc2Δ* (S1 Fig A–B), confirming that Swc2, as well as Swr1, are required for the remodeling activity of the SWR complex [38]. Overall, these results indicate that SWR is primarily responsible for the formation of most ZZ nucleosomes. However, more than half of the AZ nucleosomes persisted in *swc2Δ*, suggesting that the genomic distribution of H2A.Z in the absence of SWR activity cannot be ignored (Fig 1C).

To determine the genomic locations of H2A.Z nucleosomes in WT and *swc2Δ* cells, we isolated H2A.Z-containing nucleosomes and analyzed the associated nucleosomal DNA by sequencing as previously described [16]. Briefly, bulk chromatin was extracted from

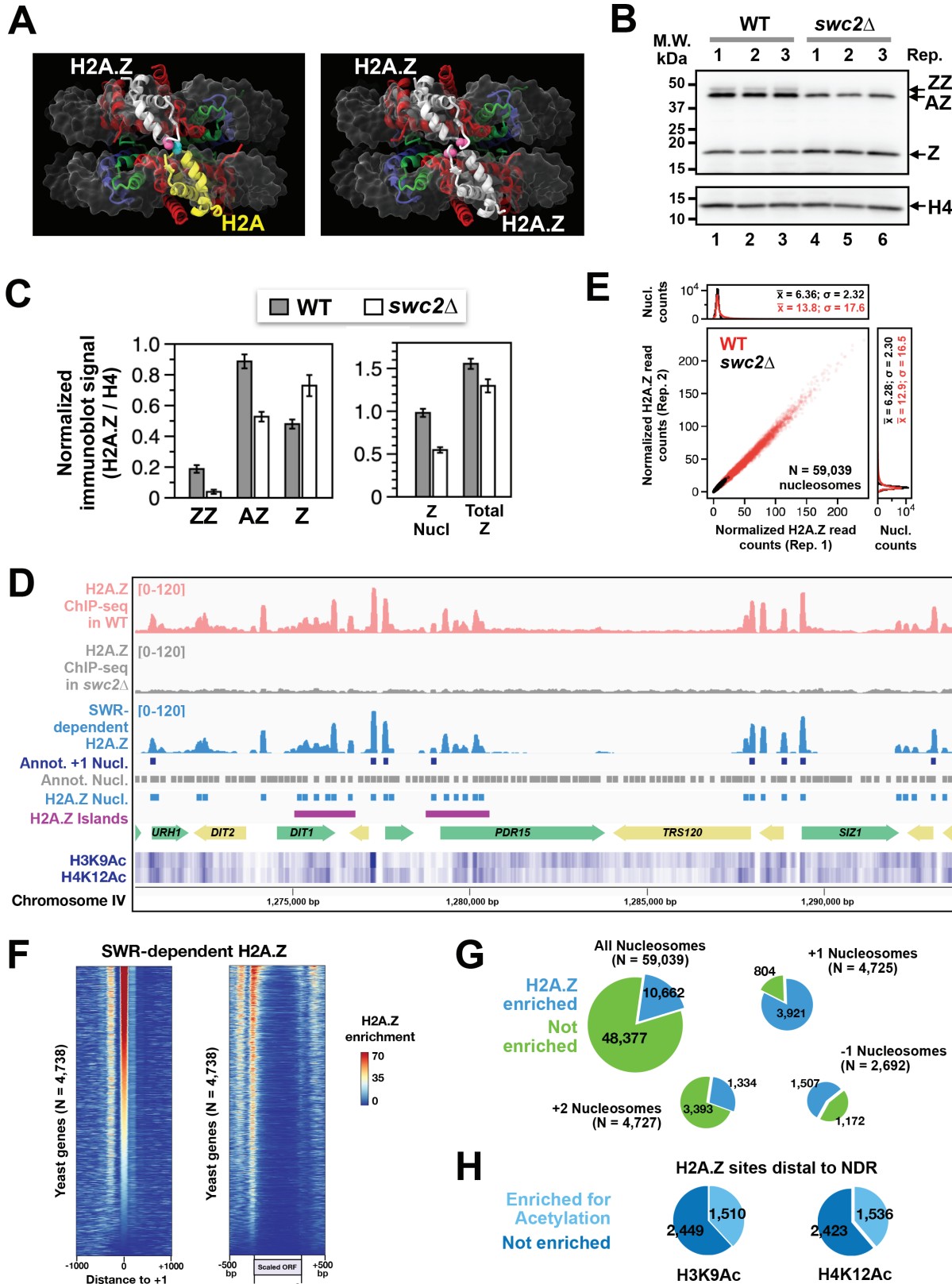

Fig 1. SWR-dependent H2A.Z-containing nucleosomes. (A) Structural representations of heterotypic H2A/H2A.Z nucleosome (left; PDB: 5B32) and homotypic H2A.Z nucleosome (right; PDB: 5B33) from humans. Pink residues correspond to T46 of yeast Htz1, and

the blue residue corresponds to N39 of yeast Hta1. **(B)** VivosX analysis of WT and *swc2Δ* cells expressing *HTZ1(T46C)-2xFLAG* and *2xV5-HTA1(N39C)* alleles. Total proteins extracts were analyzed by non-reducing SDS-PAGE and immunoblotting with anti-FLAG antibody (top) and anti-H4 antibody (bottom). Biological replicates (Rep) represent extracts from independent transformants. **(C)** Quantification of the immunoblots in B. Normalized signals represent band intensities of AZ, ZZ, and Z relative to H4. **(D)** H2A.Z ChIP-seq profiles for WT (pink) and *swc2Δ* (grey) cells. The SWR-dependent H2A.Z signal (blue) was derived by subtracting *swc2Δ* from WT. Grey blocks: annotated nucleosomes [40,41]. Dark blue blocks: +1 nucleosomes. Light blue blocks: SWR-dependent H2A.Z nucleosomes (this study). Heatmaps: H3K9Ac and H4K12Ac signals [34]. Magenta lines: H2A.Z islands. **(E)** Scatter plot showing H2A.Z ChIP-seq read counts at annotated nucleosomes between biological replicates. Histograms depicting datapoint densities are shown above and to the right of the main plot. **(F)** Heatmaps of SWR-dependent H2A.Z signals (WT minus *swc2Δ*) aligned at the center of +1 nucleosomes (left) or at the start and end of open reading frames (ORF) (right). **(G)** Relative number of identified H2A.Z-enriched sites in the indicated categories. The lower number of +1 nucleosomes in Fig 1G compared to Fig 1F is due to the exclusion of +1 sites lacking data or located in repetitive regions. **(H)** Relative number of NDR-distal H2A.Z sites enriched for H3K9 or H4K12 acetylation. The plot data for Fig 1C–1E are available in S2 Data. The plot data for the heatmaps in Fig 1F are available in S3 Data (left panel) and S4 Data (right panel). The genomic coordinates of SWR-dependent H2A.Z nucleosomes are provided in S1 Data.

spheroplasted cells and fragmented under limited micrococcal nuclease (MNase) digestion (S1 Fig C). Chromatin particles were immunoprecipitated using anti-FLAG agarose to target the 2xFLAG epitope tag on the C-terminus of H2A.Z (S1 Fig D). The immunoprecipitated DNA fragments were sequenced on the Illumina platform and mapped to the yeast genome (version R64-1-1). The resulting H2A.Z enrichment signals of WT and *swc2Δ* were initially normalized to Reads Per Million (RPM) based on read counts. To account for the reduced levels of H2A.Z-containing nucleosomes in *swc2Δ* (55.8% of WT, Fig 1C), a scaling factor of 0.558 was applied to the *swc2Δ* H2A.Z profile (Fig 1D).

In *swc2Δ*, the chromatin-bound H2A.Z was broadly distributed, with virtually no distinct H2A.Z peaks across the genome (Fig 1D and S2 Fig). This finding suggests that SWR is primarily responsible for site-specific H2A.Z deposition in yeast cells. The normalized H2A.Z counts at 59,039 annotated nucleosomes in non-repetitive regions from two biological replicates were plotted for WT (red) and *swc2Δ* (black) (Fig 1E) [40,41]. The replicates showed strong correlations ($R^2 = 0.99$ for WT; $R^2 = 0.94$ for *swc2Δ*, and we identified 10,662 nucleosomal positions with significant levels of SWR-dependent H2A.Z ($p<0.05$), using the *swc2Δ* H2A.Z ChIP-seq data to model noise as the null hypothesis (Fig 1E, and S1 Data). Consistent with previous data, SWR deposits H2A.Z predominantly at +1 nucleosomes, with 3,921 of 4,725 +1 positions showing enrichment (Fig 1F and 1G). However, a substantial amount of H2A.Z was also deposited at non-(+1) positions in a SWR-dependent manner, which was represented by subtracting the *swc2Δ* profile from the WT profile (Fig 1D and 1F). Other NDR-proximal locations, such as +2 and -1 nucleosomes, accounted for 1,334 and 1,507 H2A.Z-containing nucleosomes, respectively. These findings indicate that 3,900 H2A.Z-enriched sites (i.e., 10,662 minus 3,921, 1,334 and 1,507) distal to NDRs (defined as not +1, -1 or +2) still require SWR for deposition (Fig 1G).

In addition, we identified clusters of SWR-dependent H2A.Z sites, referred to as H2A.Z islands. Using a criterion that defined an island as containing ≥ 6 H2A.Z-containing nucleosomes within a 1,500-bp window, we identified 264 H2A.Z islands across the yeast genome (Fig 1D, S3 Fig magenta bars and S1 Data). Many nucleosomes within these H2A.Z islands are not particularly enriched for histone acetylation (S3 Fig B–C). In fact, only one third of the NDR-distal H2A.Z sites across the genome are associated with H3 acetylation at lysine 9 (N = 1,510) or H4 acetylation at lysine 12 (N = 1,536) (Fig 1H). Therefore, SWR deposits H2A.Z at thousands of distinct H2A.Z sites by an unknown targeting mechanism that is not directly linked to NDRs or histone acetylation.

## SWR preferentially inserts H2A.Z into a subset of nucleosomes endogenously enriched for H2A.Z

To uncover unknown determinants of SWR targeting, we employed an unbiased biochemical approach to interrogate the substrate specificity of SWR. This approach utilized a library of nucleosomes liberated from H2A.Z-deficient chromatin by MNase digestion. The use of canonical substrate was crucial, as nucleosomes pre-loaded with H2A.Z are poor substrates for SWR [26].

To prepare native canonical nucleosomes, we used an *htz1Δswr1Δ* strain carrying an episomal 2xV5-tagged *HHF2* gene as the sole source of histone H4 (S4 Fig A). Logarithmically growing cells were spheroplasted under conditions optimized to minimize proteolytic clipping of histone tails [42]. Nucleosomes and poly-nucleosomes were liberated from the insoluble chromatin pellet by MNase digestion (S4 Fig B–C). The nucleoproteins were then purified by anti-V5 immunoprecipitation followed by V5 peptide elution. Nucleosomes were separated from poly-nucleosomes by sucrose gradient sedimentation before being dialyzed into a buffer compatible with the remodeling reactions (S4 Fig D). The native nucleosomal substrate, along with a recombinant nucleosomal control, were incubated with native SWR, ATP and Z-B dimers that were doubly tagged with biotin and FLAG (Fig 2A–2B). Both the native and recombinant nucleosomes were active substrates of SWR, as evidenced by the slower migration of remodeled nucleosomes in native polyacrylamide gel electrophoresis (PAGE), caused by the tags (primarily FLAG) on the Z-B dimer (Fig 2A and 2C) [28].

Nucleosomes remodeled by SWR, i.e., the AZ and ZZ species, were pulled down by streptavidin-coated paramagnetic beads and eluted with dithiothreitol (DTT), as the biotin moiety was linked to H2A.Z (Htz1) via a cleavable disulfide linkage (Fig 2D–2E). As a proof of concept, a partially remodeled reaction containing a mixture of recombinant AA, AZ and ZZ nucleosomes (input) was subjected to streptavidin pulldown and DTT elution (Fig 2E, lanes 1–3). The eluate fractions were enriched for the AZ and ZZ species, with undetectable levels of AA nucleosomes, indicating the specificity of the pulldown (Fig 2E). Similarly, native nucleosomes containing one or two copies of the dual-tagged Z-B dimers from the SWR-mediated reaction were pulled down by streptavidin beads and released into the eluate fraction after DTT treatment (Fig 2E, lanes 4–6). To isolate the subset of native nucleosomes preferentially remodeled by SWR, we performed time-course reactions. Aliquots from the remodeling reactions, which require $Mg^{2+}$ as a cofactor, were quenched by the addition of ethylenediaminetetraacetic acid (EDTA) at 15, 30, and 45 min (Fig 2F). The native nucleosomes were prepared in replicates and reacted with SWR in two independent time-course experiments (Fig 2F and S5 Fig A). DNA from the eluate and flow-through (FT) fractions of the replicate reactions was purified and analyzed by paired-end sequencing (S5 Fig B).

The sequencing reads of the eluate and FT fractions were mapped to the yeast genome, and read coverages were normalized to read counts (S6 Fig). H2A.Z (Z)-enrichment was defined as the difference in read coverage between the eluate and FT fractions. Although SWR broadly inserted H2A.Z into nucleosomes across the genome, preferred nucleosomal sites were observed (S6 Fig). To minimize the influence of DNA linkers on SWR's substrate selectivity, a read size filter (140-154 bp) was applied to the sequencing data. The reproducibility of the site-specific H2A.Z deposition by SWR was assessed by comparing Z-enrichment values between experimental replicates at previously annotated, non-repetitive nucleosomal sites (N = 59,039) (Fig 3A). Positive correlations were observed for Z-enrichment values between replicates at all three time points ($R^2$: 0.49, 0.45 and 0.45 for 15, 30, and 45 min, respectively). The top 3% of Z-enriched nucleosomes (1,777) were defined as SWR-preferred sites (red) and the bottom 3% as unpreferred sites (blue) (Fig 3B). Importantly, we found the SWR-preferred

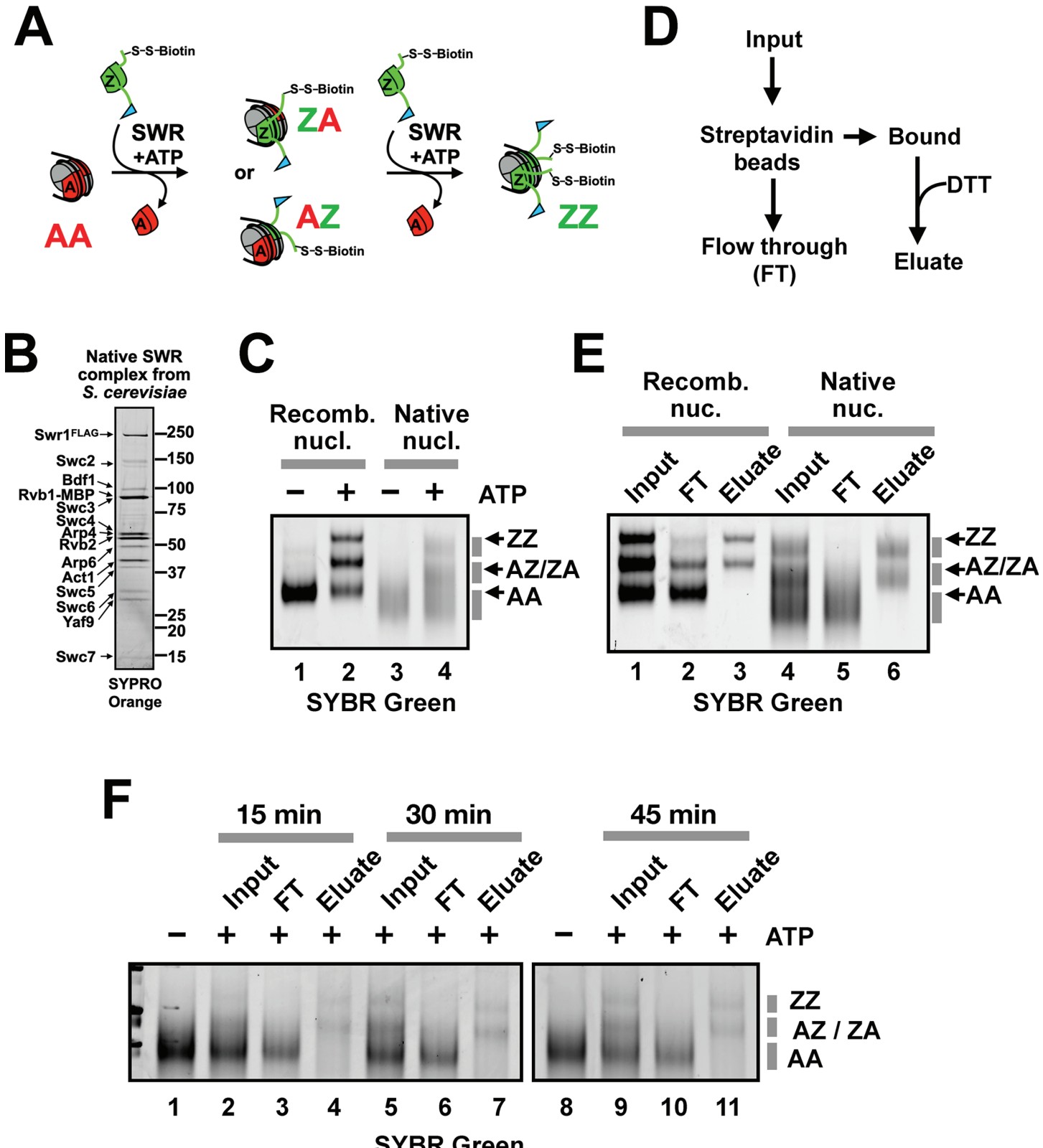

**Fig 2. SWR-mediated H2A.Z deposition reaction using native yeast nucleosomes. (A)** The histone exchange assay. Blue flag indicates the 3xFLAG tag on Htb1. The S-S-Biotin label indicates the cleavable biotin tag on Htz1(V126C). Red sectors: A-B dimers. Green sectors: Z-B dimers. Grey sectors: H3-H4 dimers. **(B)** Native SWR

complex purified by the ASAP technique was separated on an 8-16% polyacrylamide gel and analyzed by SYPRO orange staining. **(C)** Histone exchange reactions analyzed by 6% polyacrylamide / 0.5x TBE electrophoresis. Recombinant nucleosomes were used in lanes 1 and 2, native nucleosomes in lanes 3 and 4. **(D)** Strategy for isolating SWR-remodeled nucleosomes. Input: nucleosomes partially remodeled by SWR in the presence of Z-B dimers that were biotinylated on Htz1 and FLAG-tagged on Htb1. **(E)** Recombinant nucleosomes and native nucleosomes partially remodeled by SWR (input) were pulled down with streptavidin-coated beads and eluted with DTT (eluate). FT fractions represent the unbound materials of the streptavidin pulldown. Cell equivalent amounts of input, FT, and eluate were analyzed by 6% PAGE and SYBR green staining. **(F)** Same as E except that the remodeling reactions were quenched by EDTA at the indicated times before subjected to streptavidin pulldown.

sites were significantly enriched for H2A.Z in vivo (see Statistics section in Methods). At the 15-, 30-, and 45-min time points, 35%, 41%, and 45% of preferred nucleosomes (out of 1,777) were enriched for H2A.Z in vivo, respectively (Fig 3B and S1 Data). By contrast, randomly selected sites would be expected to overlap with native H2A.Z nucleosomes at a frequency of only 18% (i.e., 10,662/59,039) (Fig 1G).

To further understand how SWR selects its nucleosomal substrates in vitro, we plotted Z-enrichment values across the genome (Fig 3C and S7 Fig). Under our reaction conditions, SWR preferentially loaded H2A.Z into many, but not all, regions that are endogenously enriched for H2A.Z, including both NDR-proximal and distal sites, as well as H2A.Z islands. While a significant proportion of +1 nucleosome sites were identified (13.6%, 14.2%, and 16.2%) using the top 3% threshold, many endogenous H2A.Z +1 sites were missed (Fig 3B). To refine our prediction of SWR-preferred sites specifically at +1 nucleosomes, we performed unsupervised *k*-means clustering analysis (k = 3) using Z-enrichment values around +1 nucleosomes (S8 Fig). This analysis identified 1,718, 1,471, and 1,544 SWR-preferred +1 nucleosomes (out of 4,731) at 15, 30, and 45 minutes of reaction time, respectively (Fig 4A–4B and S1 Data). Of those, 2,401 were preferred by SWR at least once. Consistent with the genomic distribution of native H2A.Z, the +1 nucleosomes preferred by SWR (for all 3 time points) were generally enriched at intergenic regions but depleted across coding regions (Fig 4C–4D). However, SWR exhibited an artifactual preference for the 3' end of genes not observed in vivo.

## SWR prefers Poly(dA:dT) tracts on the top strand of nucleosomal DNA entry/exit sites

To decipher the sequence composition of the nucleosomal substrates preferred by SWR, we performed dinucleotide motif analysis. Previous studies have showed that native nucleosomes in yeast are preferentially positioned on sequences with alternating dinucleotide patterns that are symmetrical around the nucleosomal dyad. This symmetry is associated with how the two halves of the nucleosomal DNA coil around the histone octamer [43]. We repeated this analysis by plotting the 16 dinucleotide frequencies of 67,538 nucleosome positioning sequences annotated by Brogaard et al. [43] (S9 Fig). For comparison, the average genomic frequencies of dinucleotides were also plotted. In general, dAdA and dTdT dinucleotides are enriched at the minor grooves of the DNA facing the histone octamer, whereas dSdS dinucleotides are out of phase. Other dinucleotides, such as dAdT, dTdA, dWdS, and dSdW, exhibited alternating patterns that are symmetrical around the dyad but their patterns are more nuanced (S9 Fig).

Having established the oscillating patterns of dinucleotide frequencies across positioned nucleosomes, we focused on SWR-preferred nucleosomes (present at two or more time points in Fig 3A, N = 1,069) that overlap the Brogaard annotation to uncover specific dinucleotide motifs (Fig 5 and S10 Fig). Our analysis showed that SWR-preferred nucleosomes (dotted lines) were enriched for dAdA dinucleotides on the top strand on the left half of the nucleosomal DNA relative to genome averages (solid color) (Fig 5A). A similar pattern was independently observed on the right half of the preferred nucleosomes, reflected by the enrichment

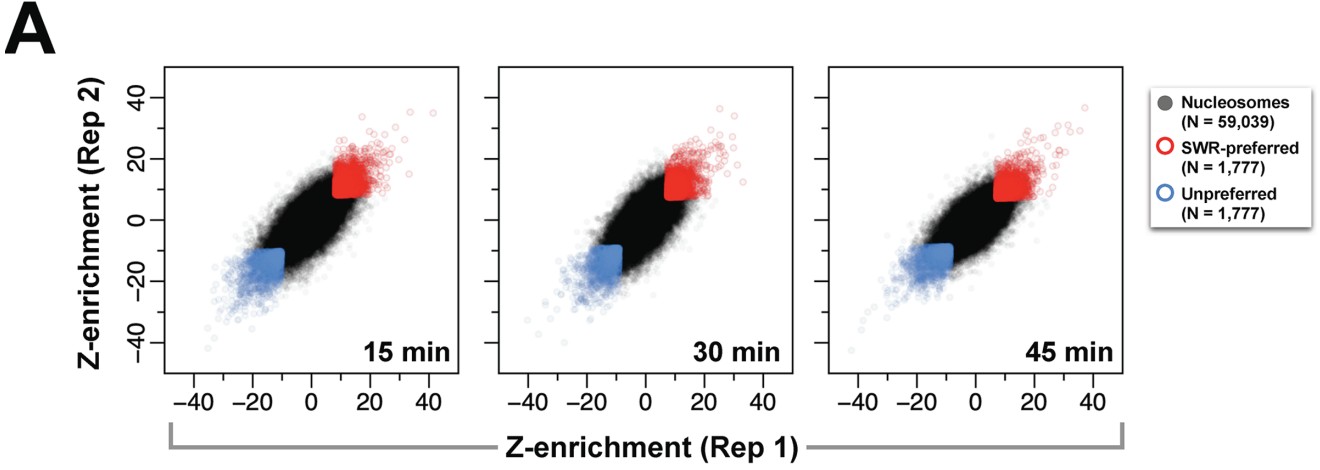

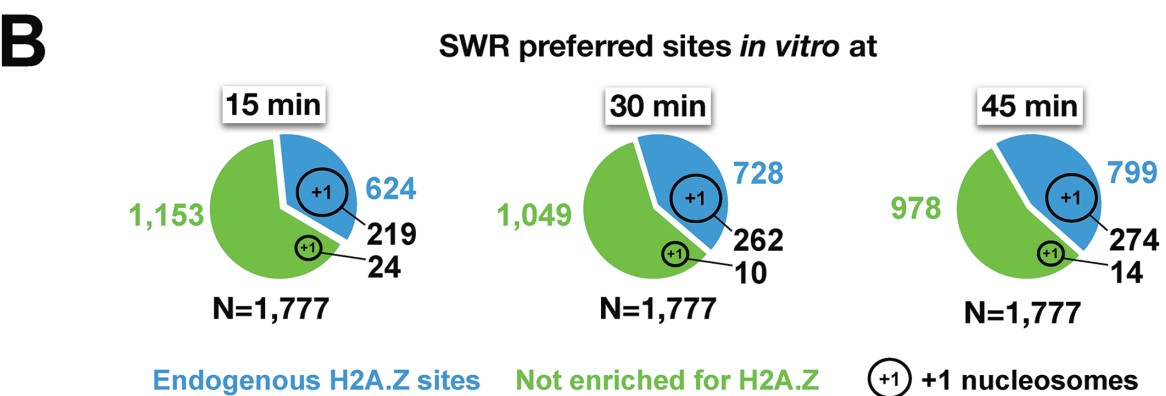

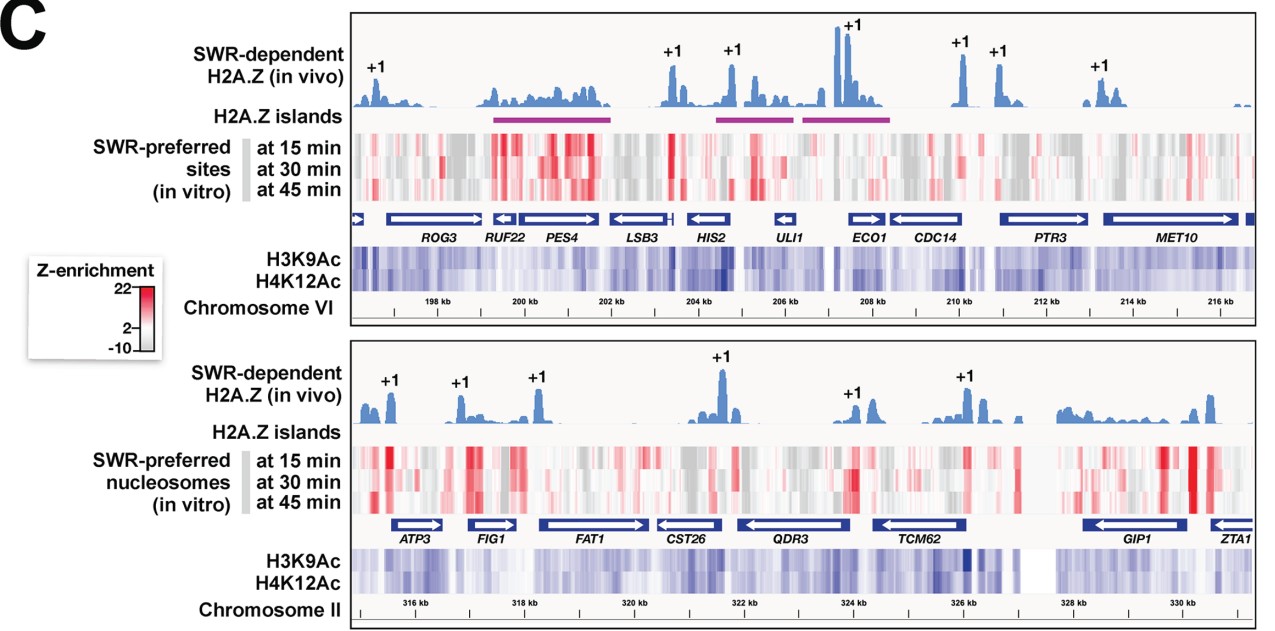

**Fig 3. Identification of SWR-preferred and unpreferred sites.** (**A**) Z-enrichment values at 59,039 non-repetitive genomic locations for the indicate time points. Red: top 3% of high Z-enrichment values. Blue: bottom 3%. (**B**) Pie charts showing the fraction of SWR-preferred sites (top 3%) that overlap

with endogenous SWR-dependent H2A.Z sites (blue) versus unenriched sites (green). Black circles (not drawn to scale) represent nucleosomes at +1 positions. **(C)** Genomic distribution of SWR-preferred sites. Z-enrichment values (red heatmaps) plotted against the endogenous SWR-dependent H2A.Z (light blue), H3K9Ac and H4K12Ac (blue heatmaps). Two representative regions were shown. Annotated +1 nucleosomes are indicated. Magenta lines: H2A.Z islands. The plot data for Fig 3A and track information for Fig 3C are available in S5 Data. The genomic coordinates of SWR-preferred nucleosomes are provided in S1 Data.

of complementary dTdT dinucleotides on the top strand (Fig 5B). These findings indicate that the dAdA and dTdT enrichments in SWR-preferred nucleosomes exhibit a palindromic pattern relative to the nucleosomal dyad. Additionally, dTdA and dAdT dinucleotides were enriched, albeit to a lesser extent, in SWR-preferred nucleosomes (Fig 5C–5D). The increase in dWdW dinucleotide frequencies in SWR-preferred substrates was offset by a decrease in dCdC, dGdG, and dGdC frequencies (Fig 5E–5G). By contrast, other dinucleotide motifs exhibited more subtle differences in the preferred substrates (Fig 5H and S10 Fig). When the dinucleotide frequencies of the unpreferred substrate (N = 1,236) were compared to those of the averaged nucleosomes, the reverse trend was generally observed: dWdW dinucleotides were depleted, while dSdS were enriched in the unpreferred substrates (Fig 5 and S10 Fig, gray lines).

To evaluate whether the preferred nucleosomes contain longer dA or dT tracts, we plotted the frequencies of five consecutive dA ($dA_5$) and dT ($dT_5$) residues (S11 Fig). Consistent with the results of the dinucleotide motif analysis, the $dA_5$ and $dT_5$ frequencies exhibited a symmetrical dA:dT enrichment profile relative to nucleosome averages. To determine how many SWR-preferred nucleosomes contain poly(dA:dT) tracts, we found that 79% of the preferred nucleosomes contain one or more $dA_5$ or $dT_5$, compared to 28% of the unpreferred nucleosomes. As a reference, 57% of nucleosomes across the genome contain one or more $dA_5$ or $dT_5$. On average, SWR-preferred nucleosomes contain 3.2 $dA_5$ or $dT_5$ sites, while unpreferred nucleosomes contain 0.46 site, and genome-wide nucleosomes contain 1.5 sites. When uninterrupted $dA_{10}$ or $dT_{10}$ motifs were considered, 9.0% of preferred nucleosomes contain these sequences, compared to 0.16% of unpreferred nucleosomes and 2.3% of genome-wide nucleosomes. These analyses further support the conclusion that SWR preferentially remodels nucleosomes containing dA:dT tracts.

## SWR-mediated H2A.Z insertion activity is stimulated by poly(dA:dT) tracts in nucleosomes

Given the enrichment of dAdA and dTdT dinucleotide motifs in SWR-preferred nucleosomes, we asked if the presence of poly(dA:dT) tracts within the nucleosomal sequence is sufficient to stimulate SWR's H2A.Z insertion activity in vitro. Using the Widom 601 nucleosome as a control substrate, we systematically introduced 10 or 13 consecutive dA:dT residues (designated as $dA_{10}$ or $dA_{13}$) into one half of the nucleosome. In the Position (Pos) 1 mutant, a $dA_{13}$ tract was placed at the DNA entry site [i.e., between the entry site and Super Helical Location (SHL) -6], Pos 2, 3, and 4 contained $dA_{10}$ tracts located between SHL-6 and -5, SHL-5 and -4, and SHL-4 and -3, respectively (Fig 6A, red bars). To ensure that SWR acted exclusively on the nucleosome half containing the poly(dA) tracts, a 2-nt gap was introduced at SWR's ATPase engagement site on the opposite half, thereby preventing histone exchange (Fig 6A and 6B, blue box) [28]. As a control, a reference substrate containing the canonical Widom sequence was included in all reactions (Fig 6C). Remodeling of both substrates were monitored by fluorescence densitometry on PAGE, with the reference and test substrates end-labeled with Alexa555 and Alexa647, respectively.

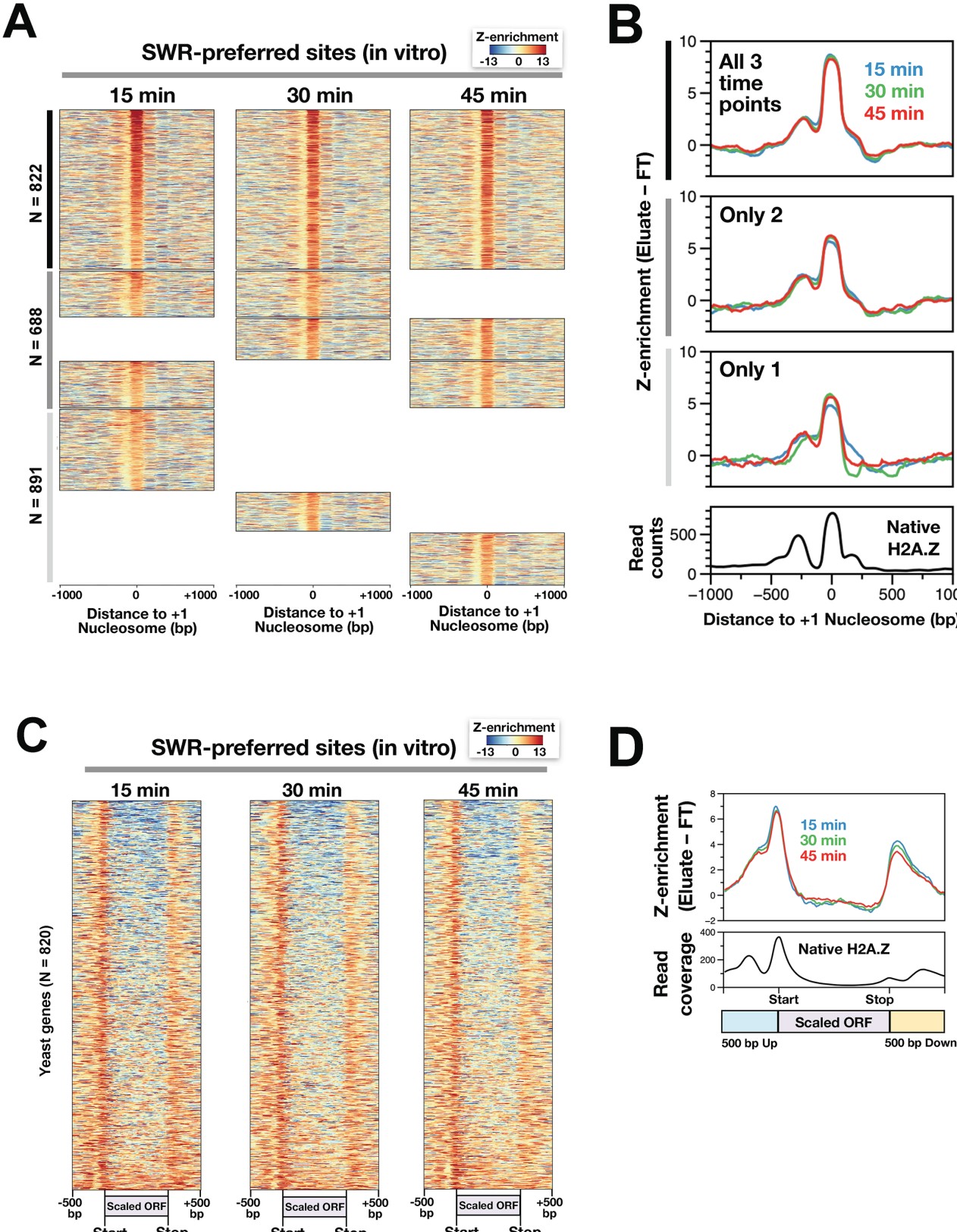

**Fig 4. SWR-preferred nucleosomes at +1 positions. (A)** Z-enrichment values of SWR-preferred +1 sites identified by *k*-means clustering analysis were aligned at +1 nucleosome dyads. Black vertical bar: +1 nucleosomes preferred by SWR at all three reaction time points. Gray bar: +1 nucleosomes preferred at two time points. Light gray bar: preferred at a single time point. **(B)** Line plots of the heatmaps in (A). Native H2A.Z

represents endogenous SWR-dependent H2A.Z enrichment, derived from the data presented in Fig 1F. **(C)** Heatmaps of Z-enrichment values along genes with SWR-preferred +1 nucleosomes. Genes were scaled to equal length and aligned at their starts and ends. **(D)** Line plots of the heatmaps in (C) versus endogenous SWR-dependent H2A.Z (black). The plot data for Fig 4 are available in S6 Data. The genomic coordinates of the sorted +1 nucleosomes are provided in S1 Data.

## SWR-preferred nucleosomes

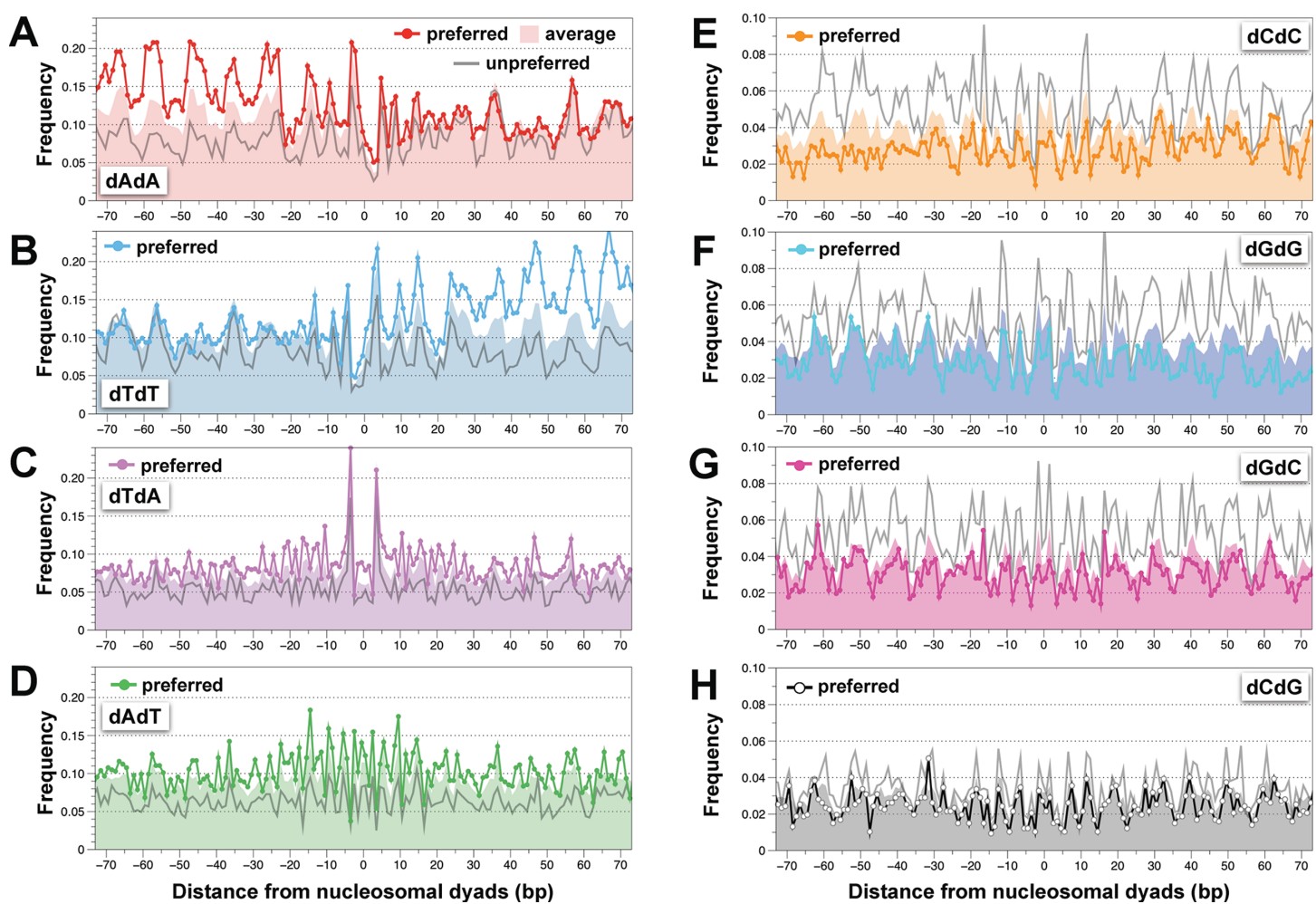

**Fig 5. Dinucleotide motif analysis of SWR-preferred and unpreferred nucleosomes. (A-H)** Dinucleotide frequencies along positioned nucleosomes are shown as dotted lines for SWR-preferred nucleosomes, gray lines for unpreferred nucleosomes, and shaded areas for genome-wide averages. Preferred and unpreferred sites were defined based on the classification in Fig 3. In total, there are 1,069 SWR-preferred nucleosomes and 1,236 unpreferred nucleosomes, while genome averages are represented by 67,538 nucleosomes. See S1 Data for genomic coordinates and sequences. The plot data for Fig 5 are available in S7 Data.

In the control reaction, where the test substrate lacked the poly(dA) tract but contained the 2-nt gap, SWR replaced the untagged nucleosomal A-B dimer on the intact side with a FLAG-tagged Z-B dimer, generating AZ nucleosomes that migrated more slowly than AA nucleosomes (Fig 6D, top). When $dA_{10}$ was introduced at Pos 3, an increased production of AZ species was observed, indicating that the poly(dA) tract can indeed stimulate SWR's H2A.Z insertion activity (Fig 6D and S12 Fig A–B, top). Notably, the stimulation was position-dependent: $dA_{10}$ at Pos 2 stimulated SWR to a lesser extent than at Pos 3, whereas $dA_{13}$ at Pos

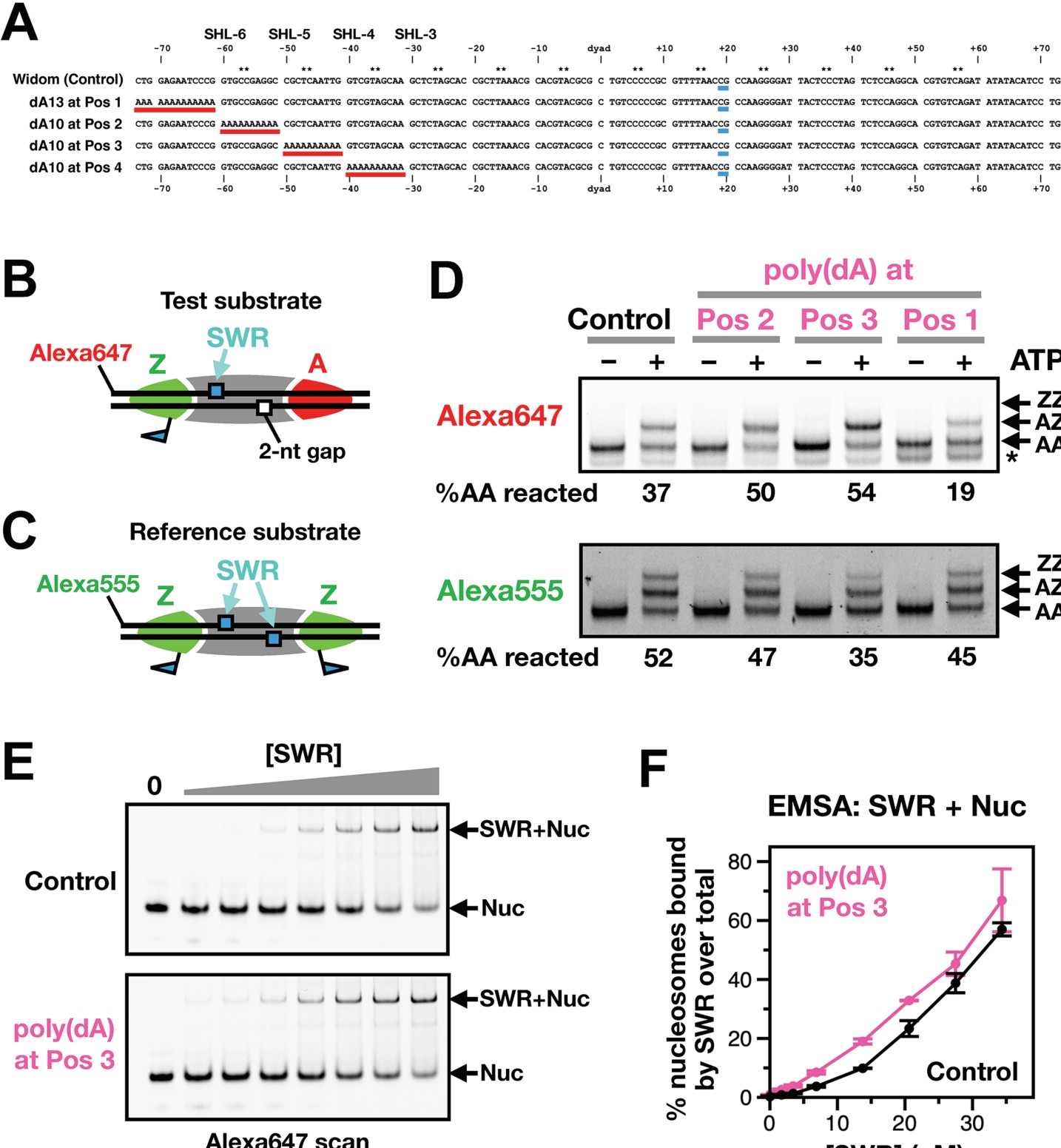

**Fig 6. Effect of poly(dA) tracts on SWR activity using reconstituted nucleosomal substrates. (A)** Alignment of the Widom 601 control sequence versus the poly(dA)-containing variants. Red bars highlighted the poly(dA) tracts in positions (Pos) 1–4. Blue bars indicated the position of the 2-nt gap. Note that the poly(dA) tracts were added to the AT-rich side of the Widom 601 sequence. **(B)** The test substrate has an Alexa647 fluorophore at the 5' end of the 'top' strand and a 2-nt gap at positions +19 to +20 nt (from the dyad) on the 'bottom' strand. **(C)** The reference substrate has the canonical Widom 601 sequence and an Alexa555 fluorophore at the 5' end.

All nucleosomal substrates are flanked by 7-bp linkers DNA on both sides. This design intentionally excludes long linker DNA, which is known to help recruit SWR, to isolate and reveal the contribution of the sequence effect on the H2A.Z deposition activity of SWR. **(D)** Fluorescent scans of the nucleosomal DNA. The expected positions for the AA, AZ and ZZ nucleosomes were indicated. An asterisk (*) indicates the expected position for hexasomes. The percentage of AA reacted is the difference in AA band intensities in the +ATP lane relative to the no ATP control. **(E)** Binding assay between SWR and nucleosomes with and without the $dA_{10}$ tract at Pos 3. Alexa 647-labeled nucleosomes (30 nM) were incubated with SWR at the indicated final concentrations for 30 min on ice before they were resolved on a 4% polyacrylamide gel containing 0.2xTB and 5% glycerol at 45V for 1.5 hr. The gel was scanned directly through the gel plates using a Typhoon 9500 imager. **(F)** Quantification of the gel shown in (E) and S12 Fig C. The line plot represents average values from the experimental replicates with error bars indicating the range of the data. The plot data for Fig 6F are available in S8 Data.

1 and $dA_{10}$ at Pos 4 inhibited SWR's activity (Fig 6D and S12 Fig A–B, top). The stimulation observed in the Pos 3 substrate was not due to an artifactually increased SWR activity, as the reference substrate in the same reaction was not enhanced but instead diminished relative to the control (Fig 6D and S12 Fig A–B, bottom).

One explanation for the increased H2A.Z deposition activity in the Pos 3 $dA_{10}$ nucleosome is that SWR binds more tightly to this substrate. To test this hypothesis, we performed a binding assay to compare SWR's affinity for the Pos 3 $dA_{10}$ nucleosomes versus the control, according to a previously described method [44]. Our results showed that SWR exhibited a higher affinity for the Pos 3 $dA_{10}$ nucleosomes (Fig 6E–6F and S12 Fig C), providing a mechanistic explanation for the stimulatory effect of the DNA sequence on SWR activity.

The nucleosomes used in the remodeling assay above had short (7-bp) linkers on both sides. To assess whether the stimulatory effect of the poly(dA) tract is also observed in substrates that mimic the +1 nucleosome adjacent to the NDR, we utilized a canonical nucleosomal substrate with a 50-bp linker on one side and a 7-bp on the other (Fig 7A). Upon incubation with SWR and epitope-tagged Z-B dimers, the control AA nucleosomes were converted into heterotypic AZ and ZA species (Fig 7B–7C), which appeared as a doublet in the native gel before progressing to the ZZ product (Fig 7D–7E). Previously, site-directed hydroxyl radical footprinting showed that the AZ species, with the Z-B dimer positioned distal to the NDR, corresponds to the lower band of the doublet, while the ZA species, with the Z-B dimer proximal to the NDR, corresponds to the upper band [28].

When a $dA_{10}$ tract was introduced between SHL-4 and -5 (i.e., Pos 3), a more prominent upper ZA band in the AZ/ZA doublet was observed, indicating that SWR exhibited a stronger insertion preference toward the NDR-proximal side of the nucleosome containing the $dA_{10}$ tract (Fig 7D–7E). Notably, this increase was accompanied by a reduced H2A.Z insertion on the NDR-distal site without the $dA_{10}$ tract relative to the control (Fig 7D–7E).

Together, these findings highlight the role of DNA sequences in modulating SWR's H2A.Z insertion activity and in guiding H2A.Z deposition across the genome.

## Discussion

### A revised model of chromatin remodeling at yeast promoters

The formation of the nucleosome-depleted promoter platform and the closely spaced nucleosome arrays within gene bodies ensure the focused assembly of the transcription preinitiation complex and accurate start site selection. The site-specific incorporation of H2A.Z by SWR at the +1 nucleosome of many genes places the variant histone at a critical juncture, where RNA Pol II transitions from initiation to elongation, contributing to accurate transcriptional response. In this study, we uncovered an underappreciated role of DNA sequences in guiding H2A.Z deposition, giving new insights into how SWR targets its histone substrate at promoters and promoter-distal regions.

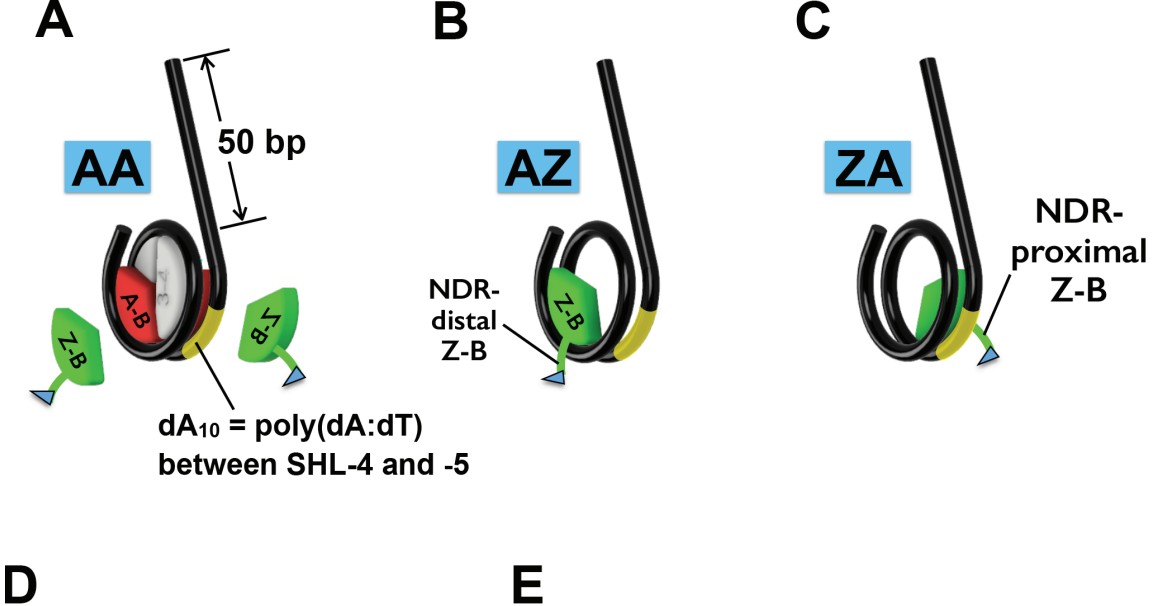

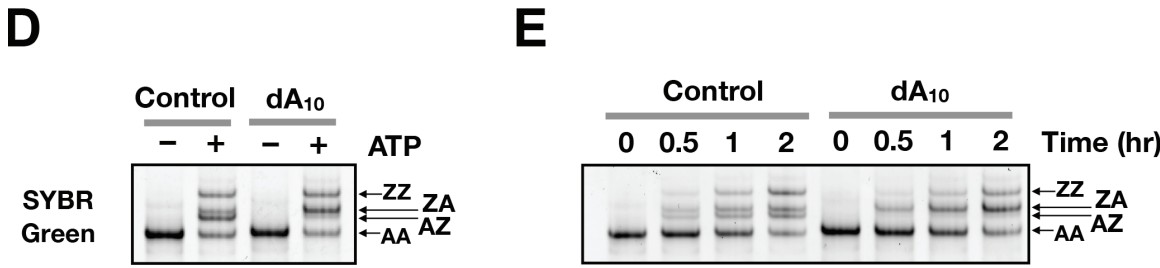

**Fig 7. SWR remodeling assay for monitoring site-specific insertion of H2A.Z into a nucleosome. (A)** Schematic representation of the asymmetric nucleosomal substrates used in the assay. These substrates feature a 50-bp linker on one side and a 7-bp linker on the other. Where indicated ($dA_{10}$), a 10-bp poly(dA:dT) tract was inserted between SHL-4 and -5 on the side closer to the long linker. Red: A-B dimers. Green: Z-B dimers, which contain a 3xFLAG tag on H2B (Htb1). Grey: H3-H4 dimers. Yellow: $dA_{10}$ tract. **(B–C)** The AZ nucleosome has the Z-B dimer inserted on the side distal to the long linker, whereas the ZA nucleosome has the Z-B dimer positioned proximal to the long linker. **(D–E)** Histone exchange reactions were allowed to proceed for 2 hours in D and for the indicated times in E. The resulting nucleosomes were resolved on a 6% polyacrylamide gel with 0.5x TBE at 110V for 2 hours, followed by SYBR Gold staining.

Yeast promoters are AT-rich, with poly(dT) and poly(dA) tracts predominantly occupying the sense strand at the upstream and downstream boundaries of the NDR, respectively (S13 Fig) [41]. DNA duplexes containing long poly(dA:dT) are poor substrates for nucleosome assembly as these homopolymeric sequences are rigid and intrinsically curled (i.e., bent and twisted) [45–48]. Thus, yeast promoter sequences are naturally depleted for nucleosomes [4]. However, DNA sequence alone is insufficient to keep 'nascent' nucleosomes away from promoters. Chromatin remodeling activities are also required to re-establish and maintain the native NDR observed in vivo [37]. The remodeler RSC, whose activity is enhanced by the poly(dA:dT) tracts, actively removes nascent nucleosomes from promoters, thereby widening the NDR (S13 Fig, step 2) [36,49]. The INO80 remodeler in turn slides the +1 nucleosome into its native position, a process that is influenced by DNA sequences (S13 Fig, step 3) [37,50]. In this study, we demonstrate that the remodeling activity of the SWR complex is also modulated by genomic sequences, with poly(dA:dT) tracts enhancing its activity. Together with the presence of NDRs and histone acetylation, DNA sequences contribute to the site-specific deposition of H2A.Z in vivo.

We propose that SWR may act upstream of RSC and INO80 to replace nucleosomal A-B dimers in nascent, promoter-proximal nucleosomes with Z-B dimers. The rationale is twofold. First, before a nascent nucleosome is slid into its native +1 position by RSC and INO80, its upstream edge is likely overlapping with the poly(dA:dT) sequence, hence placing the poly(dA) tract on the top strand of the nucleosomal DNA between SHL-4 and -6. This configuration makes the nascent nucleosome an ideal substrate for SWR-mediated histone exchange. Our data supports this model, as nucleosomes originating from the NDRs, albeit in low abundance, were preferentially remodeled by SWR (Fig 4B). Second, RSC preferentially disassembles and INO80 preferentially slides H2A.Z-containing nucleosomes over canonical nucleosomes [51,52]. Therefore, it is energetically more efficient for SWR to deposit H2A.Z before RSC and INO80 establish the native promoter architecture (S13 Fig, step 1).

## How poly(dA:dT) sequences facilitate H2A.Z insertion

At the mechanistic level, specific DNA sequences may introduce structural stress within the nucleosome, facilitating the unwrapping of nucleosomal DNA at the entry/exit sites. This structural perturbation could expand the nucleosome-free region, enhancing SWR binding, as SWR utilizes its Swc2 and Swc3 subunits to engage long stretches of exposed DNA [29]. Our in vitro binding experiment supports this model. Alternatively, but not exclusively, the rigidity of the poly(dA:dT) sequence may destabilize the nucleosomal substrate and facilitate the eviction step of the histone exchange reaction. As the Swr1 ATPase draws in the nucleosomal DNA at SHL-2 to disrupt histone-DNA contacts of the outgoing A-B dimer, the structural stress introduced by the poly(dA:dT) sequence could promote DNA unwrapping and enhance A-B dimer eviction. Why SWR prefers poly(dA:dT) tracts but not poly(dT:dA) tracts is intriguing. This suggests that poly(dA:dT) may curl the DNA in such a way that is more compatible with the transition state of the SWR-mediated reaction, whereas the curl caused by poly(dT:dA) is inverted and is less favorable.

Previously, we investigated SWR-mediated Z-B dimers deposition onto the opposite sides of an asymmetrically position nucleosome, reconstituted with the Widom sequence with a long linker (50 bp) at one end and a short linker (7 bp) at the other. Our findings revealed that Z-B dimer deposition efficiencies on the two sides depend on both reaction temperature and DNA sequences [28]. We identified a 16-bp region, spanning positions 24 to 39 bp relative to the nucleosomal dyad that, when swapped with the corresponding site on the opposite face of the nucleosome, is sufficient to bias Z-B dimer deposition towards one side at low temperatures (e.g., 4 and 10°C). The favorable sequence (5'-TAGGG AGTAA TCCCC T-3') contains two short poly(dG:dC) tracts, in contrast to the less favored sequence (5'-TCGTA GCAAG CTCTA G-3'), despite both having equivalent GC content. This observation leads to the speculation that SWR prefers poly(dG:dC) tracts. However, under our current experimental conditions at an ambient temperature of 23°C, SWR did not exhibit a preference for native nucleosomes with high dCdC and dGdG contents. The discrepancy could be attributed to the temperature at which the current SWR assays were performed. Alternatively, the preferred 16-bp region, which overlaps the A-B/Z-B dimer binding site, may contain more nuanced motifs beyond poly(dG:dC) tracts that SWR favors at low temperatures. Thus, the sequence motifs favored by SWR likely have multiple permutations that can be influenced by specific reactions conditions, with poly(dA:dT) tracts identified here representing a key element that could combine with other motifs to fine tune SWR-mediated H2A.Z insertion in vivo.

When investigating substrate preference of SWR using the native nucleosome library, SWR failed to target many +1 nucleosomes that are typically enriched for H2A.Z (Fig 3B–3C

and S7 Fig). This failure likely stems from the disruption of long linkers by MNase diges-
tion, which are presumably more important for SWR recruitment than sequence motifs at
these sites. Notably, these +1 nucleosomes are often associated with higher frequency of
poly(dG:dC) tracts, which do not favor H2A.Z insertion under our reaction conditions [28]
(Fig 5), further reinforcing the notion that a subset of endogenous H2A.Z sites do not rely
on nucleosomal sequence motifs to facilitate H2A.Z insertion. Therefore, across different
genomic locations, a complex interplay between nucleosomal linker DNA length, sequence
motifs, and histone modifications orchestrates the site-specific deposition of H2A.Z in vivo.

### Intrinsic bias of SWR against genic nucleosomes

SWR preferentially deposits H2A.Z into nucleosomes associated with intergenic over genic
sequences. One explanation is that the information-rich genic sequences are depleted for
the monopolymeric dA:dT sequences preferred by SWR. When genic regions were sub-
jected to dinucleotide motif analysis, 3-bp periodicities were observed for all 16 dinucleotide
motifs, reflecting the biased codon usage and codon-pair preferences of yeast genes (S14 Fig).
Therefore, poly(dA:dT) sequences are selected against in the coding regions, rendering these
regions unpreferred by SWR. An exception is genes encoding lysine-rich and phenylalanine-
rich domains, which include consecutive AAA and TTT codons (S15 Fig). This provides a
potential explanation for why some genes in the H2A.Z islands, such as *PES4* (Fig 3C), exhibit
non-canonical H2A.Z deposition in genic regions.

Finally, SWR exhibits a strong preference for nucleosomes at the 3' end of genes, consistent
with the presence of poly(dA) tracts (on the sense strand) associated with the transcriptional
termination signal in yeast [53]. However, unlike the preferred H2A.Z insertion at the pro-
moter regions, insertion at 3' end is non-physiological, as native H2A.Z is generally enriched
at the 5' end but not the 3' end of genes. How poly(dA:dT) sequences at terminator regions
exclude SWR-mediated H2A.Z insertion in cells is unknown and may involve masking by
termination factors in the regions [34].

In conclusion, SWR-mediated H2A.Z deposition in cells is influenced by multiple factors,
including NDRs, histone acetylation, and, as revealed by this study, specific DNA sequences.
This finding places SWR within a growing list of chromatin remodelers, including RSC,
INO80, Chd1, whose remodeling activities are tuned by DNA sequences [36,50,54]. One
emerging theme is that the genome encodes information to direct chromatin remodeling
activities or that remodelers have evolved to recognize sequence signatures associated with
distinct genomic regions, such as promoters.

This study also highlights the power of native yeast nucleosomes as substrates to interro-
gate chromatin remodeling events. When coupled with high-throughput sequencing, this bio-
chemical approach provides deeper insights into how substrate diversity influences chromatin
remodeling mechanisms.

## Materials and methods

### Yeast strains

Yeast strains used in this study are listed in S1 Table. The WT strain (yEL379) in the VivosX
analysis of Fig 1B–1C was previously described [39]. The *swc2Δ* strain (yEL575) was con-
structed by replacing the *SWC2* ORF in yEL379 with the *kanMX6* cassette using the homolo-
gous gene replacement method [55,56]. The WT strain (yEL378) in the ChIP-seq analysis was
previously reported [39]. The corresponding *swc2Δ* and *swr1Δ* variants (yEL786 and yEL1041,
respectively) were similar constructed by homologous gene replacement.

The yeast strain (yEL905), which was used to prepare native nucleosome libraries, carried an episomal *HHF2* gene with a 2xV5 tag at the N-terminus to facilitate nucleosome isolation. In addition, the strain lacked *HTZ1* and *SWR1* to remove endogenous H2A.Z. To construct yEL905, the precursor strain YYY67 was used [57]. YYY67 lacked the endogenous H3 and H4 genes, i.e., *(hht1-hhf1)Δ::LEU2 (hht2-hhf2)Δ::HIS3* and was kept alive by an *HHT1-HHF1 URA3 CEN ARS* (pMS329) plasmid. A *CEN ARS TRP1* plasmid (pEL628) containing the divergently oriented *HHT1* and *2xV5-HHF2* genes was transformed into YYY67 and was used to replace pMS329 based on the plasmid shuffle technique [58]. The resulting strain (yEL704), which expressed the *2xV5-HHF2* gene as the sole source of H4, was subjected to a second round of transformation to replace *HTZ1* with the *hphMX6* cassette [55]. Finally, the strain was subjected to a third round of transformation to replace *SWR1* with *kanMX6*. Successful removal of the *HTZ1* and *SWR1* genes was verified by colony PCR.

### Plasmids

Plasmids used in this study are listed in S2 Table. To construct the plasmid pEL628, a synthetic gene fragment of *2xV5-HHF2* (synthesized by Twist Biosciences) was subcloned into a *CEN ARS TRP1 HHT2-HHF2* plasmid (gift from Rolf Sternglanz [57]) at the *HHF2* location by digestion with *BamHI* and *NcoI* followed by recombination using the Gibson Assembly protocol (New England Biolabs).

The plasmid for glucanase production (pT22-6HisLyticase) was constructed as follows. The sequence encoding amino acid residues 37-548 of glucanase (glucan endo-1,3-beta-glucosidase from *Cellulosimicrobium cellulans*, Uniprot: P22222) was synthesized by IDT and amplified by PCR using primer pT22(NdeI)Gluc-Gibs-F and pT22(XhoI)Gluc-Gibs-R (S3 and S4 Tables). The PCR product was cloned into the pET22b vector (Novagen) pre-digested with the *NdeI* and *XhoI* restriction enzymes (Thermo) using the Gibson Assembly protocol. Sequence integrity of all plasmids was confirmed by Sanger Sequencing (Azenta).

### Proteins

**Yeast SWR complex.** Native SWR was purified from total extracts from a yeast strain (yEL427) bearing the *SWR1-3xFLAG* and *RVB1-MBP* alleles using the Another Sequential Affinity Purification (ASAP) protocol previously described [28]. The concentration of SWR was quantified by in-gel staining using the SYPRO Orange dye against a known concentration gradient of bovine serum albumin protein (Roche).

**Recombinant histone substrates.** The dual tagged Z-B dimer has a cleavable biotin tag conjugated to a cysteine residue substituted at valine position 126 (V126C) of the yeast Htz1 and a triple FLAG tag at the C-terminus of yeast Htb1 [26,28]. Both proteins were produced recombinantly using the *E. coli* strain BL21 Codon Plus (DE3)-RIL and purified according to a previous protocol, except that the FLAG-tagged Htb1 was eluted from the Q Sepharose column instead of the SP Sepharose column [59]. Biotinylation of Htz1(V126C) was carried out using N-[6-(biotinamido)hexyl]-3'-(2'-pyridyldithio)propionamide (HPDP-Biotin) (Thermo Fisher Cat.# 21341), which has a pyridyl disulfide moiety. To ensure optimal biotinylation, 48 $\mu$M of Htz1(V126C) protein was first dissolved in Buffer A [25 mM HEPES (pH 7.6), 10 mM EDTA, and 0.15 M NaCl] and pre-treated with 5 mM DTT at 37°C for 1 hr. Excess DTT was removed by gel filtration using a PD-10 column equilibrated with Buffer A. The reduced Htz1(V126C) (in 3.5 mL) was concentrated to 1 mL on an Amicon Ultra-4 (10kD MWCO) column (Millipore) before 100 $\mu$L of 4 mM HPDP-Biotin [in dimethylformamide (DMF)] was added to allow disulfide exchange at 20.8°C for 1.5 hr. To remove any unreacted HPDP-Biotin, the reaction was applied to a new PD-10 column and eluted with

3.5 mL Buffer A. The eluted protein was then unfolded by the addition of 5.02 g of guanidine-HCl and 50 $\mu$L 1M Tris-HCl (pH 7.5). Molecular grade water was added to a final volume of 7.5 mL before 2 mg of Htb1-3xFLAG protein was mixed with the biotinylated Htz1 to unfold at 20.8°C for 2 hr. Protein refolding was performed by dialysis against four changes of 2 L of Buffer B [10 mM Tris-HCl (pH 7.5), 2 M NaCl, 1 mM EDTA, 0.2 mM PMSF]. The refolded dimers were concentrated to 500 $\mu$L (using Amicon Ultra-4) and purified on a Superdex 200 Increase 10/300 GL column (Cytiva) equilibrated with Buffer B. The peak fraction was dialyzed into Buffer C [10 mM Tris-HCl (pH 7.5), 50 mM NaCl, 1 mM EDTA, 0.01% NP-40] and stored in aliquots at –80°C before ready to use in nucleosome remodeling reactions.

Recombinant nucleosomes were made by combining canonical yeast histone octamers with PCR synthesized DNA followed by refolding using the salt gradient dialysis approach previously described [28]. Recombinant yeast H2A and H2B histones were purchased from the Histone Source (CSU Fort Collins, CO) and yeast H3 and H4 were produced as described [28]. The primers and templates used to generate the Widom 601 containing fragments and the poly(dA)-containing variants are listed in S3 Table and S4 Table, respectively. The amino-modified forward primers (synthesized by IDT) were chloroform extracted twice before labeling with Alexa647- or Alexa555-conjugated ester according to the manufacturer protocols (Thermo Fisher, A20106 and A20109). The labeled primers were purified by PAGE and were used in combination with reverse primer EL338 or EL873 to amplify gBlock gene fragments (IDT), gEL172-gEL176, with or without the poly(dA) sequences. PCRs (6.076 mL each) were performed in 96-well plates using taq polymerase. PCR products were purified by phenol extraction and concentrated by ethanol precipitation as described above before injected into a Superose 6 Increase 30/100 GL column equilibrated with Buffer D [10 mM Tris-HCl (pH 7.5), 1 mM EDTA, 300 mM NaCl]. Fractions containing the labeled PCR products were concentrated to 400 $\mu$L using Amicon Ultra 4 spin columns. To introduce the 2-nt gap in the dUdU modified DNA, 100 $\mu$g of DNA was incubated with 0.1 U/$\mu$L of USER enzyme (NEB) in 500 $\mu$L at 37°C overnight. The DNA products were purified by phenol extraction and concentrated by EtOH precipitation.

**Glucanase.** Production of recombinant glucanase was performed using the *E. coli* strain BL21 DE3 RIPL transformed with the pT22-6HisLyticase plasmid. The cells were grown in 6x 500 mL of Terrific Broth supplemented with 100 $\mu$g/mL of ampicillin to log phase at OD600 0.5. Glucanase production was induced with 0.5 mM IPTG and was allowed to proceed at 25°C overnight. The cells were harvested by centrifugation, resuspended in Buffer E [50 mM Tris-HCL (pH7.6), 10 mM Imidazole, 500 mM NaCl, 3 mM beta-mecaptoethanol, and 0.2 mM PMSF] and sonicated using a microtip on a Q Sonica sonicator at 50% power for 3 min (15 sec ON, 30 sec OFF). The supernatant was clarified by centrifugation and applied to a 5-mL HisTrap HP column and eluted with a 10-400 mM linear imidazole gradient on an AKTA Pure FPLC system (Cytiva). Peak fractions were dialyzed into Buffer F (2 mM MES pH 6, 10% sucrose, 10% glycerol, 5 mM DTT) and loaded onto a 5-mL SP HP column. The protein was eluted with a linear gradient of 0-750 mM NaCl in Buffer F and analyzed by SDS-PAGE, concentrated 8-fold to 500 $\mu$L, flash frozen, and stored at –80°C.

**Native nucleosomes.** Native canonical nucleosomes were prepared from the yeast strain yEL905 that lacked the endogenous *HTZ1*, *SWR1*, *HHT1-HHF1*, and *HHT2-HHF2* genes and expressed an N-terminal 2xV5-tagged H4 from an *HHT2-(2xV5-HHF2) TRP1 CEN ARS* plasmid. Logarithmically growing cells (at 0.5 OD600) cultured in 2 L yeast extract peptone dextrose (YPD) media were harvested by centrifugation, washed, aliquoted, and flash frozen. Each cell pellet equivalent to a 400-mL of culture was thawed and washed with 6 mL of Buffer G [100 mM Tris (pH 9.4), 10 mM DTT] and incubated for 5 min at 30 min in 4 mL of Buffer H [50 mM KPO4 (pH 7.5), 0.6 M Sorbitol, 10 mM DTT] plus Inhibitor Cocktail

(InhC) [1 mM sodium fluoride, 10 mM beta-glycerophosphate, 0.5 $\mu$M trichostatin A, and 10 mM sodium butyrate and 2x Complete EDTA-free protease inhibitor mix (Roche)]. The cells were spheroplasted to 80% completion by adding 40 $\mu$L glucanase and incubated at 30°C for 15 min. Spheroplasts were washed three times with 8 mL Buffer I [50 mM HEPES (pH 7.5), 80 mM KCl, 2.5 mM MgCl2, 0.4 M Sorbitol, and InhC] and resuspended in 500 $\mu$L Buffer J [50 mM HEPES (pH 7.6), 80 mM NaCl, 0.25% Triton X-100, and InhC]. The spheroplasts were disrupted in a pre-chilled 7-mL Dounce homogenizer with a tight piston at 4°C. The crude chromatin was pelleted by centrifugation at 13,000 x g for 10 min, washed three times with 500 $\mu$L Buffer J and resuspended in 500 $\mu$L Buffer J. Chromatin fragmentation was initiated by adding 0.1 mM CaCl2 and 0.3 U/$\mu$L MNase (Worthington; Cat.# NC9391488). The digestion was allowed to proceed at 37°C for 5 min followed by quenching with 10 mM EDTA. The supernatant (chromatin solution), which contained the liberated nucleoprotein fragments, was clarified by centrifugation at 20,400xg for 5 min and filtration using an Ultrafree-MC spin column (Millipore, UFC30GV0S).

To pulldown nucleoproteins containing the 2xV5-H4 histone, the chromatin solution from two 400-mL equivalent of cultures were pooled (800 $\mu$L total) and diluted to 8 mL using Buffer K [25 mM HEPES (pH 7.6), 1 mM EDTA, 80 mM KCl, and InhC] and incubated with 200 $\mu$L (slurry volume) of anti-V5 beads (Sigma) at 4°C for 4 hr. Elution was performed using Buffer L [25mM HEPES (pH 7.6), 0.3M KCl, 1mM EDTA, 0.1% NP-40, 0.1% DOC, and 0.5 $\mu$g/$\mu$L V5 peptides] at 4°C overnight. The eluate was concentrated on a Amicon Ultra-4 centrifugal filter unit and sedimented through a 4.7-mL 15-40% sucrose gradient in 25 mM HEPES (pH7.6), 0.5 mM EDTA, 0.01% NP-40 at 45,000 RPM (366,613 xg) for 20 hr at 4°C using a Beckman SW55Ti rotor. The gradient was fractioned by pipetting and analyzed by electrophoresis in 1.3% agarose/0.5x TBE and SYBR Green I staining. Fraction 8 (top fraction is 1) represent the peak fraction of mononucleosomes and was collected and dialyzed against two changes of 1 L Buffer C.

## Biochemical assays

**VivosX.** In vivo crosslinking analysis was performed according to a previous study [39]. Briefly, yeast cells were grown in the complete synthetic media (CSM) to an optical density at 600 nm (OD600) of 0.5 before 5-mL aliquots were treated with 180 $\mu$M 4-DPS (Sigma; Cat.# 143057) for 20 min at 30°C. The cells were fixed on ice by adding 20% trichloroacetic acid (TCA) for >5 min. Fixed cells were collected by centrifugation at 2,851 xg for 5 min and washed once with 20% TCA. The cells were lysed in 400 $\mu$L 20% TCA with $\sim$ 400$\mu$L 0.7-mm Zirconia beads (Biospec Products) using a FastPrep-24 homogenizer (2x 30 sec at 6.0 M/s speed) (MP Biomedicals). The insoluble material contained the crosslinked histones and were collected by centrifugation at 20,400xg for 15 min at 4°C. The pellet was washed with 1 mL acetone. To extract proteins from the pellet, 200 $\mu$L of the TUNES-G Buffer [100 mM Tris pH 7.2, 6 M urea, 10 mM EDTA, 1% SDS, 0.4 M NaCl, 10% glycerol] + 50 mM N-ethylmalemide (Sigma; Cat.# E1271-5G) was added and the dispersed precipitates were vortexed for 1 hr at 30°C. The solubilized proteins were collected by centrifugation at 20,400xg for 10 min at 4°C and analyzed on a non-reducing 8-16% polyacrylamide gel. Immunoblotting analysis was performed by transferring the proteins to a PVDF membrane and probing with the anti-FLAG (Sigma; Cat.# F3165) or anti-H4 (Active Motif; Cat.# 91295) primary antibodies (at 1:1000) and anti-mouse HRP (at 1:2000). Immunoblots were developed using the Amersham ECL Prime detection reagent (Cytiva; Cat.# 45001216) and imaged on a LAS4010 CCD camera (Cytiva).

**ChIP-seq.** Chromatin was extracted from the yeast strains yEL378 (*SWC2 HTZ1-2xFLAG*) and yEL786 (*swc2Δ HTZ1-2xFLAG*) as described with minor modifications [16]. Briefly, yeast cells (uncrosslinked) equivalent to a 400-mL culture at 0.5 OD600 were spheroplasted and lysed in 1x pellet volume of Buffer M [same as Buffer J except that InhC50 was replaced by 2x Complete Protease inhibitors (EDTA-free, Roche; Cat.# 5056489001)] using a Dounce homogenizer. After centrifugation, the chromatin-enriched pellet was washed three times with Buffer M before it was resuspended in Buffer M supplemented with 1 mM CaCl2. The crude chromatin suspension was digested with 1 U/$\mu$L MNase for 20 min. The solubilized nucleoproteins were cleared by centrifugation at 20,400xg for 5 min at 4°C followed by filtration using an Ultrafree-MC spin column (Millipore; Cat.# UFC30GV0S).

ChIP-seq was performed as previous described but with some modifications [16]. The solubilized nucleoproteins were diluted 10 folds in Buffer N [same as Buffer K except that InhC was replaced by 1x protease inhibitor cocktail (PI; 0.34 mg/mL PMSF, 0.66 mg/mL benzamidine hydrochloride, 2.74 $\mu$g/mL pepstatin A, 0.56 $\mu$g/mL leupeptin, and 4 $\mu$g/mL chymostatin)] before incubated with 200 $\mu$L (slurry volume) of anti-FLAG M2 affinity gel (Sigma; Cat.# A2220) per 5-mL diluted nucleoproteins. Binding was performed at 4°C for 4 hr on a rotator. The FLAG beads were washed with Buffer O [25 mM HEPES (pH 7.6), 1 mM EDTA, 0.01% NP-40, 0.3 M KCl, 1x PI] and eluted with 0.5 $\mu$g/$\mu$L 3xFLAG peptides (Biopeptide) in Buffer P [25mM HEPES (pH 7.6), 1mM EDTA, 0.1% NP-40, 0.3M KCl, 0.1% DOC, 1x PI] overnight at 4°C. Nucleosomal DNA was extracted by incubating with 0.34 $\mu$g/$\mu$L Proteinase K (Invitrogen; Cat.# 25530049) in the Buffer Q [340 mM NaCl, 8.5 mM EDTA, 0.43% SDS, 0.1 ug/uL Glycogen] at 55°C for 1 hr followed by extraction with phenol:chloroform:IAA (25:24:1) and ethanol/sodium acetate precipitation. The nucleic acid pellet was resuspended in TE buffer and treated with 0.05 $\mu$g/$\mu$L RNase (Roche; Cat.# 11119915001) for >15 hr at 37°C. The DNA was purified using the QIAquick PCR purification kit (Qiagen; Cat.# 28104) and it was prepared for Illumina sequencing using the NEBNext Ultra II A Library Prep Kit (NEB; Cat.# E7645L).

**SWR remodeling reactions.** Histone exchange reactions were performed by mixing the following components. For a 25-$\mu$L reaction, 11.4 $\mu$L of Buffer R [25 mM HEPES-KOH (pH 7.6), 0.5 mM EGTA, 0.1 mM EDTA, 5 mM MgCl2, 0.17 $\mu$g/$\mu$L BSA, 50 mM NaCl, 10% glycerol and 0.02% NP-40] was mixed with the biotinylated, FLAG-tagged Z-B dimers (to 25 nM in 25 $\mu$L) in a 1.5-mL LoBind tube (Eppendorf). Then canonical nucleosomes (from native or recombinant source) were added (to 29 nM in 25 $\mu$L) followed by the addition of SWR (to 1.5 nM in 25 $\mu$L). Buffer C was added to adjust the reaction volume to 20 $\mu$L. The reaction was initiated by adding 5 $\mu$L of 1 mM ATP in Buffer S [25 mM HEPES-KOH (pH 7.6), 0.5 mM EGTA, 0.1 mM EDTA, 0.02% NP-40] or 5 $\mu$L of Buffer S (no ATP control). For time-course reactions, the mixtures were incubated at 23°C on a PCR machine for the indicated times and quenched by adding 5 $\mu$L of 10 mM EDTA. For native PAGE analysis, 5 $\mu$L of the reaction was mixed with 1 $\mu$L of Buffer T [70.5% w/v sucrose in 10 mM Tris-HCl (pH 7.8), 1 mM EDTA] and 1 $\mu$L 20 $\mu$g/$\mu$L lambda phage DNA (NEB, Cat.# N3011S). The samples were separated on a 6% polyacrylamide (0.5x TBE, 1.5 mM thickness) at 110 V for 2 hr using a Novex electrophoresis system. Gels were scanned directly on a Typhoon FLA9500 imager (Cytiva) to detect fluorescently labeled nucleosomes or stained with SYBR Green I (Thermo) prior to scanning.

To isolate the biotinylated product of SWR, histone exchange reactions were scaled up to 100 $\mu$L for each pull down reaction. Streptavidin-coated M-280 Dynabeads (Thermo Fisher; Cat.#112.50) (20 $\mu$L slurry volume) prewashed with Buffer C in a 1.5-mL LoBind tube (Eppendorf) were incubated with 80 $\mu$L of the reaction. The remaining 20 $\mu$L of the reaction was set aside as the 'input'. The mixture was incubated at room temperature for 30 min

with gentle 'flicking' of the tube every 5 min. The streptavidin-coated beads were pulled down with a magnet. The supernatant was set aside as the FT fraction. The beads were washed once with 500 $\mu$L of Buffer C and eluted with 80 $\mu$L of Buffer C supplemented with 1 mM DTT at room temperature for 30 min with gentle flicking every 5 min. Five microliters of the input, FT, and eluate fractions were mixed with 1 $\mu$L of Buffer T and analyzed by PAGE and SYBR Green staining as described above. The remaining sample of the FT and Eluate fractions were analyzed by next-generation sequencing using the Illumina platform.

To purify the nucleosomal DNA for sequencing, the FT and Eluate fractions (90 $\mu$L each) were mixed with 10 $\mu$L 3 M sodium acetate and applied to a QIAquick spin column (Qiagen). Purified DNA was visualized on a 1.3% agarose / 0.5x TBE gel and SYBR Green staining. DNA concentrations were determined using the Qubit High Sensitivity (HS) double-stranded DNA assay (Invitrogen) on a Synergy 2 plate reader (Biotek).

**SWR binding assay with nucleosomes.** For the SWR binding experiment, SWR was first concentrated to a stock concentration of 172 nM using an Amicon Ultra-0.5 mL column. The remodeler was then serially diluted using Buffer U [25 mM HEPES (pH 7.6), 1 mM EDTA, 30% glycerol, 0.01% NP-40, and 300 mM NaCl]. Next, 1 $\mu$L of diluted SWR was added to 4 $\mu$L of nucleosomal substrate, resulting in the indicated final SWR concentrations. The nucleosomal substrates had 7-bp linkers on both ends and an Alexa647 fluorophore attached on one end. The nucleosomes (30 nM) were incubated with SWR on ice for 30 min before separation by PAGE.

To prepare the gel for separating SWR-bound nucleosomes from free nucleosomes, 10 mL of 4% polyacrylamide (with an acrylamide:bis-acrylamide ratio of 69:1) containing 0.2x TB buffer (17.8 mM Tris base and 17.1 mM Boric acid) and 5% glycerol was mixed with 40 $\mu$L 10% APS and 2.6 $\mu$L TEMED to initiate polymerization. The gel was cast in a mini gel cassette by sandwiching two Mini-PROTEAN plates with 1 mM-spacers (BioRad; Cat.# 1653311) and sealed with Tyvek seam tape (Dupont). The reaction mixture (5 $\mu$L) was mixed with 1 $\mu$L of Buffer T being loaded onto the gel. Electrophoresis was performed at 45V for 1.5 hours. After electrophoresis, the gel was scanned directly through the glass plates using Alexa647 excitation and emission settings on the Typhoon imager.

## Sequencing and informatics

Sequencing libraries were prepared using the NEBNext Ultra II DNA library prep kit according to the manufacturer protocol (NEB). Sequencing libraries were pooled and analyzed on a NextSeq 550 instrument using paired-end mode with 32 cycles on both ends. Sequencing data (FASTQ files) were aligned to the *S. cerevisiae* S288C genome (version R64-1-1) using Bowtie2 with the following options: –local, –very-sensitive-local, –no-unal, –no-mixed, –no-discordant, and –phred33. For H2A.Z ChIP-seq in Fig 1D–1F, S2 Fig, and S3 Fig, as well as the initial analysis of SWR preferred substrates in S6 Fig, a sizing filter of –I 120 –X 170 was used. However, SWR exhibited a preference for nucleosomes with longer linkers (S16 Fig), consistent with previous findings [30]. To minimize the contribution of linker length on SWR's substrate preference in the in vitro assay, a more stringent filter (–I 140 –X 154) was used (S16 Fig, right). Note that the initial analysis in S6 Fig used the original filter (–I 120 –X 170 filter). Mapped reads were filtered using samtools-view with options –q 30 –F 0x4. Read coverage were calculated using bamCoverage in deepTools with options –binSize 1 –extendReads –effectiveGenomeSize 11756462 –scaleFactor X –blackListFileName Mask_v2.bed (which indicated repetitive sequences in the BED format) (S1 Data). Read coverages for H2A.Z ChIP-seq data were first normalized to Read Per Million (RPM) for both WT *SWC2* and *swc2Δ* samples. To account for the reduced chromatin-bound H2A.Z levels

in *swc2Δ* relative to WT, as determined by VivosX, the read coverage for *swc2Δ* was further scaled by a factor of 0.558. Read coverages for the eluate and FT fractions from the in vitro SWR reactions were normalized to RPM. Scaling factors (X) for all samples are listed in S1 Data. Quality control plots for nucleosome positions around +1 nucleosomes in WT, *swc2Δ*, and *htz1Δswr1Δ* are shown in S17 Fig A and C, indicating no changes in nucleosome positioning. Similarly, nucleosome positioning in the eluate and FT fractions from the SWR preference assay were unchanged (S17 Fig B), indicating that SWR does not preferentially remodel mispositioned nucleosomes. Custom Python scripts were used for nucleotide motif counting and frequency calculation. The scripts, along with usage instructions, have been deposited in the Zenodo database. See the Data Availability section for details.

## Statistics

To identify SWR-dependent H2A.Z nucleosomes, normalized H2A.Z read counts were computed at annotated nucleosomes within non-repetitive regions (N = 59,039) for both WT and *swc2Δ* samples. The difference in H2A.Z signals between WT and *swc2Δ* represents SWR-dependent H2A.Z. Using a predefined threshold of 10 (i.e., WT-*swc2Δ* ≥ 10 for both replicates), we identified a total of 10,662 SWR-dependent nucleosomal H2A.Z sites. To assess the significance of H2A.Z signals at these sites, a one-sample Z-test was performed, positing the null hypothesis that these signals represent background noise. Utilizing the normally distributed H2A.Z ChIP-seq data from the *swc2Δ* strain as a noise model, Z-scores and the associated p-values were computed under the assumption of the null hypothesis. The maximum p-values observed among the 10,662 H2A.Z sites in WT replicate 1 was 0.015, and in WT replicate 2, it was 0.028. These outcomes led to the rejection of the null hypothesis, indicating that the 10,662 SWR-dependent H2A.Z sites exhibit a significant enrichment of H2A.Z.

To determine if our in vitro assay is better at identifying endogenous H2A.Z sites (alternative hypothesis) than randomly picking nucleosomes (null hypothesis), a one-sample proportion Z-test was performed. Using the 15-min time point as an example, the success rate in identifying H2A.Z nucleosomes in our assay is 35% ($\hat{p}$) relative to 18% (p) in the population (N = 59,039). Given a sample size of 1,777, the Standard Error (SE) is 0.0091. The Z-score, calculated as $z = (\hat{p} - p) / SE$, for our in vitro assay performance is therefore [(0.35 - 0.18) / 0.0091] ≈ 17.6. This results in a p-value of virtually zero, indicating that our in vitro assay (for the 15 min time point) is significantly better at identifying endogenous H2A.Z sites than random selection. Our 30-min and 45-min time points, with identification rates ($\hat{p}$) of 41% and 45%, respectively, are therefore more significant than the 15-min time point at identifying H2A.Z sites.

To assess if the top 3% of SWR-preferred nucleosomes are more frequently associated with +1 nucleosomes than expected by chance, a one-sample proportion Z-test was performed. At the 15-min time point, 243 out of 1,777 SWR-preferred nucleosomes were associated with +1 nucleosomes, corresponding to a success rate of 13.7% ($\hat{p}$). This is higher than the expected random association rate of 8.02% (p) if +1 nucleosomes (4,731 out of 59,039) were picked at random. Using an SE of 0.0064, the resulting Z-score is 8.66 for the 15 min time point. Similarly, the Z-scores for the 30- and 45-min time points are 11.3 and 12.7, respectively, further supporting that SWR-preferred nucleosomes are significantly associated with +1 sites.

## Supporting information

**S1 Fig. Quality control analyses for the H2A.Z ChIP-seq experiment. (A)** VivosX analysis was performed using WT, *swc2Δ* and *swr1Δ* strains bearing the *HTZ1(T46C)-2xFLAG* and

*2*x*V5-HTA1(N39C)* alleles. Three independent cultures (indicated as Rep 1-3) for each strain were grown and analyzed by non-reducing SDS-PAGE and anti-FLAG immunoblotting. **(B)** Quantification of the immunoblot in A. A t-test was performed to evaluate whether the levels of H2A.Z differed between *swr1Δ* and *swc2Δ*. The p-values indicate that there is no significant difference between them. **(C)** Control for MNase digestion. Chromatin before (-) and after (+) MNase treatment from WT and *swc2Δ* cells was analyzed by agarose gel electrophoresis and SYBR Gold staining. T: DNA extracted from total extracts. S: DNA extracted from soluble fractions, which were used in the anti-FLAG pulldown reactions against Htz1-2xFLAG. **(D)** Control for IP efficiency. Equivalent amounts of soluble chromatin before (input) and after immunoprecipitation (FT) were analyzed by SDS-PAGE and anti-FLAG immunoblotting. Rep: replicate. The plot data for S1 Fig B are available in S9 Data.
(PDF)

**S2 Fig. Distribution of H2A.Z at non-repetitive nucleosomal sites in WT and *swc2Δ*cells.** **(A)** Normalized H2A.Z read counts centered around the dyads of 59,044 nucleosomes in non-repetitive regions (black) for WT (left) and *swc2Δ* (right) cells. Profiles show nucleosomes grouped by H2A.Z levels; endogenously enriched H2A.Z sites in red and non-H2A.Z sites in blue. **(B)** Heatmaps show the alignment of these 59,044 non-repetitive nucleosomes around the nucleosomal dyads used in panel A, for both WT and *swc2Δ*cells. The plot data for S2 Fig A are available in S10 Data. The plot data for the heatmaps in S2 Fig B are available in S11 Data (left panel) and S12 Data (right panel) as gzip-compressed text files.
(PDF)

**S3 Fig. H2A.Z islands.** **(A)** Same as 1D except that two other regions with H2A.Z islands are shown. **(B)** A heatmap showing SWR-dependent H2A.Z levels of 264 H2A.Z islands. H2A.Z islands were aligned at their starts and ends, scaled to equal length, and sorted by H2A.Z levels. **(C)** Heatmaps showing H3K9Ac and H4K12Ac levels of the 264 H2A.Z islands. **(D)** Averaged SWR-dependent H2A.Z (endogenous) along 264 H2A.Z islands were compared to Z-enrichment scores representing SWR's preference in vitro. The track information for S3 Fig A and plot data for S3 Fig B–D are available in S13 Data.
(PDF)

**S4 Fig. Quality control analyses of native nucleosomes.** **(A)** Complementation test showing that the *2xV5-HHF2* gene is functional. The *HHT2-(2*x*V5-HHF2) TRP1 CEN ARS* plasmid, the untagged control or the empty vector were transformed into a yeast strain that lacked the endogenous H3 and H4 genes but was kept alive by a wild-type *HHT1-HHF1 URA3 CEN ARS* plasmid. Ten-fold serially diluted cells (starting at 1 OD600) were spotted onto synthetic complete media lacking uracil and tryptophan (left) or media supplemented with 5-FOA (right). **(B)** Optimization of MNase digestion. Yeast chromatin was incubated with MNase for the indicated times before extracted for DNA analysis. **(C)** Scale-up of MNase digestion before (–) and after (+) MNase treatment. **(D)** Nucleosomes containing V5-tagged H4 were affinity purified using anti-V5 agarose and sedimented through a 15-40% sucrose gradient. The fractions were analyzed by 1.3% agarose / 0.5x TBE electrophoresis and SYBR gold staining.
(PDF)

**S5 Fig. SWR reaction time course with native nucleosomes.** **(A)** Replicate 2 of the reaction time course as in Fig 2F. **(B)** The nucleosomal DNA from the indicated fractions were

extracted and analyzed on a 1.3% agarose/0.5x TBE gel and stained with SYBR green. Sequencing libraries were prepared from the DNA in the FT and eluate fractions.
(PDF)

**S6 Fig. Examination of SWR preferred sites.** Sequencing read coverages of the nucleosomes in the eluate (SWR-preferred) and FT (unpreferred) fractions of the streptavidin pulldown after histone exchange reactions. Red: eluate. Blue: FT. Black traces: endogenous H2A.Z. Three representative regions are shown. The plot data for S6 Fig are available in S14 Data.
(PDF)

**S7 Fig. Genomic distribution of SWR-preferred sites.** Same as Fig 3C, with additional regions included in the plot. The track information for S7 Fig is available in S15 Data.
(PDF)

**S8 Fig. k-mean clustering analysis.** Z-enrichment values were plotted around 4,731 annotated +1 nucleosomes. Their profiles were analyzed by *k*-means cluster (*k* = 3). The resulting +1 nucleosome list can be found in S1 Data. The plot data are available in S16 Data.
(PDF)

**S9 Fig. Dinucleotide motif analysis. (A)** Dinucleotide frequencies of dAdA, dTdT, dCdC, and dGdG plotted along nucleosome positioning sequences (N = 67,538) centered at nucleosomal dyads plus 20 bp of flanking regions. The box on the right shows the frequencies of the dinucleotides across the genome. **(B-D)** Same as A, except showing the indicated dinucleotide frequencies. The plot data for S9 Fig are available in S17 Data.
(PDF)

**S10 Fig. Dinucleotide motif analysis of SWR-preferred and unpreferred nucleosomes. (A–H)** Same as Fig 5, except showing other dinucleotides. The plot data for S10 Fig are available in S18 Data.
(PDF)

**S11 Fig. Poly(dA:dT) tract frequencies of SWR-preferred and unpreferred nucleosomes. (A–B)** Same as Fig 5, except showing the frequency of five consecutive dA ($dA_5$) and dT ($dT_5$). The plot data for S11 Fig are available in S19 Data.
(PDF)

**S12 Fig. Effects poly(dA) tracts on SWR activity. (A)** A replicate of the experiment in Fig 6D. **(B)** Same as A, except including a nucleosomal substrate with a poly(dA) tract at Pos 4. **(C)** A replicate of the binding experiment in Fig 6E.
(PDF)

**S13 Fig. An updated model for the formation of the promoter platform.** Pink circles: NDR-proximal nucleosomes containing H2A. Green circles: NDR-proximal nucleosomes containing H2A.Z. Grey circles: NDR-distal nucleosomes. AAA and TTT indicates poly(dA) and poly(dT) tracts respectively. Blue crab: SWR complex. Orange crab: INO80 complex. Trapezoid: RSC complex.
(PDF)

**S14 Fig. Dinucleotide motif analysis of genic regions. (A–D)** The coding regions of 6,401 yeast genes were divided into 21-bp tiled fragments in-frame with the genetic codes. The

indicated dinucleotide frequencies were averaged over 422,795 tiled regions. The plot data for S14 Fig are available in S20 Data.
(PDF)

**S15 Fig. Codon frequencies in SWR-preferred and unpreferred regions.** The plot data for S15 Fig are available in S21 Data.
(PDF)

**S16 Fig. Read length distribution** Read length distribution of the nucleosomal DNA in the eluate and FT fractions of the in vitro SWR-mediated reactions were analyzed after applying the indicated sizing filters. The plot data for S16 Fig are available in S22 Data.
(PDF)

**S17 Fig. Nucleosome positioning analysis. (A)** MNase-seq analysis of WT and *htz1Δswr1Δ* mutant. Sequencing read counts were centered at the dyads of 4,738 annotated +1 nucleosomes. **(B)** Sequencing reads from the eluate and FT fractions of the in vitro SWR-mediated reactions (after Streptavidin pulldown) were mapped to 822 SWR-preferred +1 nucleosome sites identified through *k*-means analysis in Fig 4. **(C)** Input sequencing reads from WT and *swc2Δ* used in the H2A.Z IP reactions were centered at the dyads of +1 nucleosomes. Rep: biological replicate. The plot data for S17 Fig are available in S23 Data.
(PDF)

**S1 Table. Yeast strains.**
(PDF)

**S2 Table. Plasmids.**
(PDF)

**S3 Table. Primer sequences.**
(PDF)

**S4 Table. Gene Block DNA sequences.**
(PDF)

**S1 Data. Genomic Data Files:** BED coordinates for: (1) SWR-dependent H2A.Z nucleosomes in vivo, (2) all nucleosomes based on Ref. [40,41], (3) H2A.Z islands, (4-6) nucleosomes preferred by SWR at 15 min, 30 min, 45 min reaction times, (7) nucleosomes preferred by SWR at +1 nucleosomes used in Fig 5B, (8) positioned nucleosomes [43] and their associated DNA sequences, identified as preferred or unpreferred by SWR, (9) regions used to remove repetitive sequences, and (10) scaling factors for sequencing data.
(XLSX)

**S2 Data. Plot Data for Fig 1C–1E.**
(XLSX)

**S3 Data. Plot Data for Fig 1F (left panel).**
(TXT)

**S4 Data. Plot Data for Fig 1F (right panel).**
(TXT)

**S5 Data. Plot Data for Fig 3A and track information for Fig 3C.**
(XLSX)

**S6 Data. Plot Data for Fig 4.**
(XLSX)

**S7 Data. Plot Data for Fig 5.**
(XLSX)

**S8 Data. Plot Data for Fig 6F.**
(XLSX)

**S9 Data. Plot Data for S1 Fig B.**
(XLSX)

**S10 Data. Plot Data for S2 Fig A.**
(XLSX)

**S11 Data. Plot Data for S2 Fig B (left panel).**
(GZIP–TXT)

**S12 Data. Plot Data for S2 Fig B (right panel).**
(GZIP–TXT)

**S13 Data. Track information for S3 Fig A and Plot Data for S3 Fig B–D.**
(XLSX)

**S14 Data. Plot Data for S6 Fig.**
(XLSX)

**S15 Data. Track information for S7 Fig.**
(XLSX)

**S16 Data. Plot Data for S8 Fig.**
(XLSX)

**S17 Data. Plot Data for S9 Fig.**
(XLSX)

**S18 Data. Plot Data for S10 Fig.**
(XLSX)

**S19 Data. Plot Data for S11 Fig.**
(XLSX)

**S20 Data. Plot Data for S14 Fig.**
(XLSX)

**S21 Data. Plot Data for S15 Fig.**
(XLSX)

**S22 Data. Plot Data for S16 Fig.**
(XLSX)

**S23 Data. Plot Data for S17 Fig.**
(XLSX)

**S1 Raw images. Raw images for Fig 1B, 2B, 2C, 2E, 2F, 6D, 6E, 7D, 7E, S1A, S1C, S1D, S4B, S4C, S4D, S5A, S5B, S12A, S12B, and S12C.**
(ZIP)

## Acknowledgments

We thank Nancy Hollingsworth [Stony Brook University (SBU)], Ashby Morrison (Stanford University), Dongyan Tan (SBU), Christopher Brownlee (SBU), and Jason Harper (SBU) for helpful advice and Rolf Sternglanz (SBU) for providing the yeast strain YYY67 and the plasmid pMS329.

## Author contributions

**Conceptualization:** Cynthia Converso, Leonidas Pierrakeas, Ed Luk.

**Data curation:** Cynthia Converso, Leonidas Pierrakeas, Ed Luk.

**Formal analysis:** Cynthia Converso, Leonidas Pierrakeas, Ed Luk.

**Funding acquisition:** John M Denu, Ed Luk.

**Investigation:** Cynthia Converso, Leonidas Pierrakeas, Lirong Chan, Shalvi Chowdhury, Emily de Onis, Ed Luk.

**Methodology:** Cynthia Converso, Leonidas Pierrakeas, Lirong Chan, Shalvi Chowdhury, Emily de Onis, Vyacheslav I Kuznetsov, John M Denu, Ed Luk.

**Project administration:** Leonidas Pierrakeas, Ed Luk.

**Resources:** Vyacheslav I Kuznetsov, Ed Luk.

**Software:** Ed Luk.

**Supervision:** Leonidas Pierrakeas, Ed Luk.

**Validation:** Cynthia Converso, Leonidas Pierrakeas, Ed Luk.

**Visualization:** Ed Luk.

**Writing – original draft:** Cynthia Converso, Ed Luk.

**Writing – review & editing:** Cynthia Converso, Leonidas Pierrakeas, Ed Luk.

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
