## [Editor Report · Decision Letter 0]

26 Apr 2024

Dear Dr Luk,

Thank you for submitting your manuscript entitled "Impact of Nucleosomal DNA Sequences on SWR Remodeler Activity and Histone H2A.Z Targeting" for consideration as a Research Article by PLOS Biology.

Your manuscript has now been evaluated by the PLOS Biology editorial staff, as well as by an academic editor with relevant expertise, and I am writing to let you know that we would like to send your submission out for external peer review.

Once your full submission is complete, your paper will undergo a series of checks in preparation for peer review. After your manuscript has passed the checks it will be sent out for review. To provide the metadata for your submission, please Login to Editorial Manager (https://www.editorialmanager.com/pbiology) within two working days, i.e. by Apr 28 2024 11:59PM.

Kind regards,

Richard

Richard Hodge, PhD

rhodge@plos.org

PLOS

---

## [Decision Letter · Decision Letter 1]

16 Aug 2024

Dear Ed,

Thank you for your patience while we considered your revised manuscript "Impact of Nucleosomal DNA Sequences on SWR Remodeler Activity and Histone H2A.Z Targeting" for publication as a Research Article at PLOS Biology. Please accept my sincere apologies for the long delays that you have experienced during the peer review process. Your revised study has been evaluated by the PLOS Biology editors, the Academic Editor and two independent reviewers.

Please note that we contacted Nature Communications to obtain the previous reviewer reports and their identities, but unfortunately, we were unable to obtain the names of the previous reviewers in this case. As a result, we recruited two additional reviewers to arbitrate the responses to the previous reviews.

As you will see, the new reviewers think that revised manuscript satisfactorily addresses many of the comments provided in the reports at Nature Communications. However, Reviewer #1 notes that the manuscript does not provide important data to clarify whether the effect of the DNA sequence is to increase the affinity of SWR for nucleosomes and/or impacts the enzymatic activity. The reviewer suggests several experiments to tease this apart, some of which were requested in the previous round. This includes changing the ratios of the enzymatic reaction or using gel shift/pull down experiments with nucleosomes that do/do not have the polyadenine tracts. In addition, Reviewer #2 notes that several important controls are missing for the data and raises concerns with some of the conclusions and interpretations of the data.

In light of the reviews, we would like to invite you to revise the work to thoroughly address the reviewers' reports. Given the extent of revision needed, we cannot make a decision about publication until we have seen the revised manuscript and your response to the reviewers' comments. Your revised manuscript is likely to be sent for further evaluation by all or a subset of the reviewers.

**IMPORTANT - SUBMITTING YOUR REVISION**

*Re-submission Checklist*

*Published Peer Review*

*PLOS Data Policy*

*Blot and Gel Data Policy*

Kind regards,

Richard

Richard Hodge, PhD

rhodge@plos.org

REVIEWS:

Reviewer #1: The authors had done a good job of addressing many points raised previously, but some key concerns remain that have not been adequately answered. The question remains whether the DNA sequence effect observed for SWR complex removing H2A and replacing it with H2A.Z is due to increased affinity of SWR for nucleosomes and/or impacts the enzymatic activity. The requested experiments appear to be straight forward such as changing the enzymatic reactions from having 20X excess nucleosomes where SWR affinity can be a determining factor to where SWR1 is equal molar or in slight excess to ensure SWR's affinity is not a key factor. Given the title of the paper with its potential implications (i.e. targeting) and the importance of this determination, it seems imperative to perform the necessary experiments. Is there any experimental reason why these experiments cannot be performed? Even though these experiments do not address other questions which the authors point out such as what step in the enzymatic reaction; whether the rate at which the dimer is exchanged, the other dimer is inserted or the rate of ATP hydrolysis, these experiments are nonetheless essential for this paper and are the minimal that is required.

The other approach to address this question is to perform gel shift experiments or pull down assays with nucleosomes that have or don't have the poly dA-dT tracts as pointed out by the authors. It should be sufficient to perform these experiments with only the recombinant nucleosomes and not have to go the level of doing the same experiments with the native nucleosomes combined with whole genome sequencing. In this way it should be feasible for the authors to perform these limited number of experiments and avoid the concerns expressed by the authors of this being too expansive for the current paper. I don't feel this is beyond the scope of the paper but is rather critical information that is missing for the story to be considered complete.

The authors' suggestion it is OK to not consider the effect of linker DNA on the DNA sequence specificity of the SWR complex because SWR can still do dimer exchange with nucleosomes that have no or only short linker DNA seems incomplete and potentially misleading. It is possible that the DNA sequence specificity they observe with nucleosomes lacking sufficient linker DNA may not be observed when nucleosomes have sufficient linker DNA length. In the event this is the case then it would show this DNA sequence specificity is very likely not biologically relevant. We agree with the authors' comments about the potential for a complex interplay between linker DNA, DNA sequence specificity and histone modifications, which in its totality is beyond the scope of this paper. The linker DNA experiments however only need be done with the model mononucleosome substrates as already performed with the simple change of adding on linker DNA and not have to perform the myriad of other potential experiments that could be done towards this general purpose. It doesn't seem like a good rationale to separate out the impact of linker DNA from the DNA sequence of the core nucleosome as suggested in the paper given the likely interplay between the two.

Reviewer #2: The goal of this study is to characterize the DNA sequence determinants that stimulate the H2A/H2A.Z exchange activity of the SWR complex and its targeting to specific nucleosomes in the yeast genome (namely +1 nucleosomes at the transcription start site of most yeast genes). The question is relevant to the field of chromatin biology and the experiments are well designed with a clever combination of in vivo and in vitro approaches. They conclude that the "preferred" nucleosome substrates for SWR activity contain poly A tracts upstream of the dyad axis and propose a model in which SWR acts on nucleosomes initially assembled in the NDR or the promoter region before remodeling steps mediated by RSC and Ino80 that re-position these nucleosomes to their "canonical" +1 position.

While the model seems plausible based on their results, there is a lack of crucial controls, which coupled with some puzzling reasoning undermines the robustness of their conclusions and their interpretation of the data as I explain below.

1. There are logical flaws in the interpretation of their otherwise ingenious and informative in vivo crosslinking approach from Figure 1B. Just because H2A.Z incorporation is reduced by ~50% in swc2D mutants compared to wt, it does not mean that H2A.Z is incorporated by SWR independent mechanisms in 50% of all H2A.Z nucleosomes in wt cells. It just means that H2A.Z incorporation in the absence of SWR activity is possible but incorporation efficiency is two fold lower than in wt where SWR activity is normal. This assay does not exclude the possibility that H2A.Z incorporation is SWR dependent for all H2A.Z conatining nucleosomes in WT cells.

2. Despite their claims to the contrary, that conclusion (that H2A.Z is incorporated by SWR independent mechanisms in 50% of all H2A.Z nucleosomes in wt cells) is not corroborated by their H2A.Z ChIP-seq experiments in wt and swc2D cells (Figure 1 E-G).

Even though they do not explain very well how they normalized the data, I'm assuming each dataset (wt and swc2D) was normalized separately to its respective total read count (ppm) and then "corrected" for the differences in H2A.Z incorporation between wt and swc2D by dividing the swc2D signal by 0.558 (or multiplying the wt signal by the same factor). Incidentally, the latter correction is quite unnecessary since the first normalization to total read count already gives enrichment peaks relative to the rest of the genome for each dataset. This correction does not actually allow for a direct comparison between equivalent genomic loci in each dataset, because each dataset was internally normalized first. One can only make direct comparisons between two different datasets if they are externally normalized to the same "spike-in" control.

Despite this confusing normalization method, they do however reach the correct conclusion that, since H2A.Z enrichment levels in swc2D are indistinguishable from background, SWR is responsible for all H2A.Z incorporation into specific nucleosome locations (+1, -1, what they call H2A.Z islands etc..). Unfortunately, they then go a step too far and over interpret their data by claiming that 50% of wt H2A.Z nucleosomes do not depend on SWR activity. Their data only suggests that H2A.Z is incorporated randomly in the absence of SWR activity. There is however no evidence in their data that this kind of random SWR-independent H2A.Z incorporation also occurs in wt cells where SWR activity is normal. Precisely because the H2A.Z ChIP signal in swc2D is indistinguishable from the background, it is impossible to tell what proportion of that signal actually comes from bona fide H2A.Z nucleosomes and what proportion comes from non-specific antibody binding to random epitopes. The latter would be all the more likely if H2A.Z incorporation is reduced genome-wide, as they show in Figure 1B.This is why ChIP-seq data are, as a rule, normalized to input, which had apparently not been done for this analysis. Consequently, the identification of SWR dependent H2A.Z nucleosomes in wt cells by subtraction of swc2D data from wt data amounts to the same as normalization of wt data to its own input. The second is actually more accurate since nucleosome positions in wt and swc2D may vary (another control that is not provided). Nevertheless, I do agree that the H2A.Z nucleosomes that they identified as SWR dependent in wt are indeed dependent on SWR activity but they should really use inputs for normalization, eliminate unnecessary normalization "corrections" and take out their conclusion about SWR-independent H2A.Z incorporation in wt cells.

3. Their strategy from here on is to identify nucleosomal DNA sequence features that favor SWR activity with an in vitro assay in which SWR incorporates H2A.Z into native mononucleosomes derived from htzD/swrD cells. While the strategy is reasonable, there are two important controls that are missing and without which their results are difficult to interpret correctly.

The results in Figure 5B-C and S8B show a puzzling enrichment of H2A.Z nucleosomes in regions that are depleted of nucleosomes in vivo: NDRs in promoters and at the 3' end of genes. As pointed out by reviewer 1 (from the previous review of this article), these sequences should be depleted from their set of native mononucleosomal substrates if a) their MNase digestion degraded all internucleosomal DNA and left only mononucleosomes with very short 6/7 bp linkers, and b) nucleosome positioning in their htzDswrD strain from which they derived their mononucleosomal set is identical to the wt nucleosome positioning of their native H2A.Z reference dataset.

To control for a) and exclude the possibility that +1 nucleosomes from their native mononucleosome set are favored by SWR just because they have longer linkers, they need to provide fragment size distribution of their paired-end reads, as was suggested by reviewer 1. Incidentally their response to reviewer 1 does not address the problem: just because one expects that AT rich regions will be preferentially cut by MNAse does not mean that was actually the case in their mononucleosome set (MNAse digestions are finicky and it's quite easy to over or under-digest), especially since judging from the smear in lanes 4-6 of Figure 3B they almost certainly isolated mononucleosomes with linkers of variable sizes. To control for b) they need to provide nucleosome positioning data for their htzDswrD strain, again something that was suggested by reviewer 1 and was not addressed in their response. Indeed, a potential increase in nucleosome occupancy in the NDRs coupled with "fuzzy" positioning in ORFs in that strain would essentially enrich for NDR sequences in their substrate set, in which case the apparent preference of SWR for polyA rich NDR regions could just be due to an over-representation in their substrate set of what would have been wt NDR sequences but are actually nucleosomal sequences in their htzDswrD mutant. It is impossible to make any meaningful conclusions about DNA sequence preferences of SWR targets without these two controls.

4. The analysis in Figure 6 shows a modest 25% enrichment in dAdA frequencies in the first half of what they call SWR "preferred" nucleosomes, which represent only 3% of all isolated H2A.Z nucleosomes and have at best only a twofold enrichment over flow through (see Figure 4 and Figure 3). I would advise the authors to be very cautious when they draw general conclusions on SWR activity based on these modest sequence preferences. The analysis shows oscillations in dAdA frequencies over the first half of a composite nucleosome based on 1700 nucleosome sequences. It is not clear how many if any of these nucleosomes actually have 10 A or 13 A tracts in the first half of their DNA sequence. The test constructs from Figure 7 with 10 A and 13 A tracts at different positions within the first half of the nucleosome are therefore hardly representative of any one nucleosome from the "preferred" SWR set. While they do show that test substrates with a polyA tract at positions 2 and 3 are modestly "preferred" by SWR over a control substrate (35% to 46%), a preference probably due to the inherent instability of nucleosomes with this type of sequence, it is difficult to assess the in vivo frequency of nucleosome formation over these kinds of sequences and consequently estimate how general the model in Figure 8 may be, if it happens at all.

---

## [Decision Letter · Decision Letter 2]

22 Jan 2025

Dear Ed,

Thank you for your patience while we considered your revised manuscript "Impact of Nucleosomal DNA Sequences on SWR Remodeler Activity and Histone H2A.Z Targeting" for publication as a Research Article at PLOS Biology. This revised version of your manuscript has been evaluated by the PLOS Biology editors, the Academic Editor and the original reviewers.

Based on the reviews, I am pleased to say that we are likely to accept this manuscript for publication, provided you satisfactorily address the following data and other policy-related requests that I have provided below (A-G):

(A) We routinely suggest changes to titles to ensure maximum accessibility for a broad, non-specialist readership. In this case, we would suggest a minor edit to the title, as follows. Please ensure you change both the manuscript file and the online submission system, as they need to match for final acceptance:

“H2A.Z deposition by the SWR complex is stimulated by polyadenine DNA sequences in nucleosomes”

(B) You may be aware of the PLOS Data Policy, which requires that all data be made available without restriction: http://journals.plos.org/plosbiology/s/data-availability. For more information, please also see this editorial: http://dx.doi.org/10.1371/journal.pbio.1001797

-Supplementary files (e.g., excel). Please ensure that all data files are uploaded as 'Supporting Information' and are invariably referred to (in the manuscript, figure legends, and the Description field when uploading your files) using the following format verbatim: S1 Data, S2 Data, etc. Multiple panels of a single or even several figures can be included as multiple sheets in one excel file that is saved using exactly the following convention: S1_Data.xlsx (using an underscore).

-Deposition in a publicly available repository. Please also provide the accession code or a reviewer link so that we may view your data before publication.

Figure 1C-D, 1E-F, 3A, 3C, 4A-D, 5A-H, S1B, S2A-B, S3A-D, S6, S7, S8, S9A-D, S10A-H, S11, S14A-D, S15, S16, S17A-C

(C) Thank you for already providing the raw sequencing data in the NIH SRA and GEO databases. Please forgive me if I am mistaken, but I wasn’t sure if the ChIP-seq data had been deposited here as well? If not, I would be grateful if you could please include this data in the deposition.

(D) Please also ensure that each of the relevant figure legends in your manuscript include information on *WHERE THE UNDERLYING DATA CAN BE FOUND*, and ensure your supplemental data file/s has a legend.

(E) We require the original, uncropped and minimally adjusted images supporting all blot and gel results reported in the following Figures:

Figure 1B, 2B-C, 2E-F, 6D-F, 7D-E, S1A, S1C-D, S4B-D, S5A-B, S12A-C

We will require these files before a manuscript can be accepted so please prepare and upload them now. Please carefully read our guidelines for how to prepare and upload this data: https://journals.plos.org/plosbiology/s/figures#loc-blot-and-gel-reporting-requirements

(F) Please ensure that your Data Statement in the submission system accurately describes where your data can be found and is in final format, as it will be published as written there.

(G) Per journal policy, if you have generated any custom code during the course of this investigation, please make it available without restrictions. Please ensure that the code is sufficiently well documented and reusable, and that your Data Statement in the Editorial Manager submission system accurately describes where your code can be found.

We expect to receive your revised manuscript within two weeks.

*Published Peer Review History*

*Press*

Kind regards,

Richard

Richard Hodge, PhD

rhodge@plos.org

Reviewer reports:

Reviewer #1 (Blaine Bartholomew, signs review): The authors have addressed all of my concerns.

Reviewer #2: The reviewers have adequately addressed all the points that I raised in my review and I am happy to recommend the manuscript for publication.

---

## [Editor Report · Decision Letter 3]

7 Feb 2025

Dear Ed,

On behalf of my colleagues and the Academic Editor, Tom Misteli, I am pleased to say that we can accept your manuscript for publication, provided you address any remaining formatting and reporting issues. These will be detailed in an email you should receive within 2-3 business days from our colleagues in the journal operations team; no action is required from you until then. Please note that we will not be able to formally accept your manuscript and schedule it for publication until you have completed any requested changes.

In addition, I am pleased to say that we have commissioned a Primer article to accompany your Research Article. I will be back in contact with you in a few weeks so you have the chance to review the article once it is submitted.

PRESS

Best wishes, 

Richard

Richard Hodge, PhD

rhodge@plos.org

PLOS
